# SCAN: Learning Hierarchical Compositional Visual Concepts

**Irina Higgins, Nicolas Sonnerat, Loic Matthey, Arka Pal,**
**Christopher P Burgess, Matko Bošnjak, Murray Shanahan,**
**Matthew Botvinick, Demis Hassabis, Alexander Lerchner**
DeepMind, London, UK
`{irinah,sonnerat,lmatthey,arkap,cpburgess,`
`matko,botvinick,demishassabis,lerchner}@google.com`

## Abstract

The seemingly infinite diversity of the natural world arises from a relatively small set of coherent rules, such as the laws of physics or chemistry. We conjecture that these rules give rise to regularities that can be discovered through primarily unsupervised experiences and represented as abstract concepts. If such representations are compositional and hierarchical, they can be recombined into an exponentially large set of new concepts. This paper describes SCAN (Symbol-Concept Association Network), a new framework for learning such abstractions in the visual domain. SCAN learns concepts through fast symbol association, grounding them in disentangled visual primitives that are discovered in an unsupervised manner. Unlike state of the art multimodal generative model baselines, our approach requires very few pairings between symbols and images and makes no assumptions about the form of symbol representations. Once trained, SCAN is capable of multimodal bi-directional inference, generating a diverse set of image samples from symbolic descriptions and vice versa. It also allows for traversal and manipulation of the implicit hierarchy of visual concepts through symbolic instructions and learnt logical recombination operations. Such manipulations enable SCAN to break away from its training data distribution and imagine novel visual concepts through symbolically instructed recombination of previously learnt concepts.

## 1 Introduction

State of the art deep learning approaches to machine learning have achieved impressive results in many problem domains, including classification (He et al., 2016; Szegedy et al., 2015), density modelling (Gregor et al., 2015; Oord et al., 2016a;b), and reinforcement learning (Mnih et al., 2015; 2016; Jaderberg et al., 2017; Silver et al., 2016). They are still, however, far from possessing many traits characteristic of human intelligence. Such deep learning techniques tend to be overly data hungry, often rely on significant human supervision and tend to overfit to the training data distribution (Lake et al., 2016; Garnelo et al., 2016). An important step towards bridging the gap between human and artificial intelligence is endowing algorithms with compositional concepts (Lake et al., 2016; Garnelo et al., 2016). Compositionality allows for reuse of a finite set of primitives (addressing the data efficiency and human supervision issues) across many scenarios by recombining them to produce an exponentially large number of novel yet coherent and potentially useful concepts (addressing the overfitting problem). Compositionality is at the core of such human abilities as creativity, imagination and language-based communication.

We propose that concepts are abstractions over a set of primitives. For example, consider a toy hierarchy of visual concepts shown in Fig. 1. Each node in this hierarchy is defined as a subset of visual primitives that make up the scene in the input image. These visual primitives might include factors like object identity, object colour, floor colour and wall colour. As one traverses the hierarchy from the subordinate over basic to superordinate levels of abstraction (Rosch, 1978) (i.e. from the more specific to the more general concepts corresponding to the same visual scene), the number of concept-defining visual primitives decreases. Hence, each parent concept in such a hierarchy is an

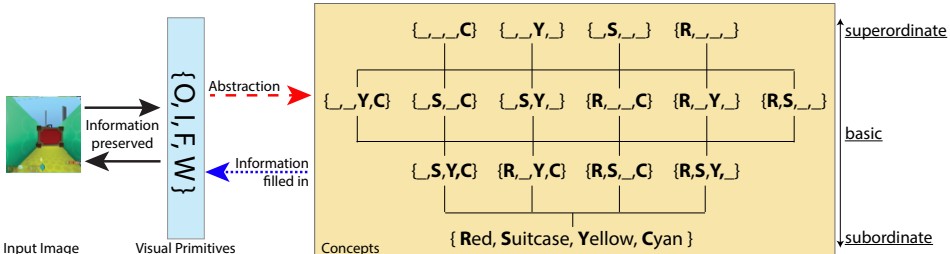

Figure 1: Schematic of an implicit concept hierarchy built upon a subset of four visual primitives: object identity ($I$), object colour ($O$), floor colour ($F$) and wall colour ($W$) (other visual primitives necessary to generate the scene are ignored in this example). Concepts form an implicit hierarchy, where each parent is an abstraction over its children and over the original set of visual primitives (the values of the concept-defining sets of visual primitives are indicated by the bold capital letters). In order to generate an image that corresponds to a concept, one has to fill in values for the factors that got abstracted away (represented as "_"), e.g. by sampling from their respective priors. Given certain nodes in the concept hierarchy, one can traverse the other nodes using logical operations. See Sec.3 for our formal definition of concepts.

abstraction (i.e. a subset) over its children and over the original set of visual primitives. A more formal definition of concepts is provided in Sec. 3.

Intelligent agents are able to discover and learn abstract compositional concepts using little supervision (Baillargeon, 1987; Spelke, 1990; Baillargeon, 2004; Smith & Vul, 2013). Think of human word learning – we acquire the meaning of words through a combination of a continual stream of unsupervised visual data occasionally paired with a corresponding word label. This paper describes SCAN (Symbol-Concept Association Network, see Fig. 2A), a neural network model capable of learning grounded visual concepts in a largely unsupervised manner through fast symbol association. First, we use the $\beta$-VAE (Higgins et al., 2017a) to learn a set of independent representational primitives through unsupervised exposure to the visual data. This is equivalent to learning a disentangled (factorised and interpretable) representation of the independent ground truth "generative factors" of the data (Bengio et al., 2013). Next, we allow SCAN to discover meaningful abstractions over these disentangled primitives by exposing it to a small number of symbol-image pairs that apply to a particular concept (e.g. a few example images of an apple paired with the symbol "apple"). SCAN learns the meaning of the concept by identifying the set of visual primitives that all the visual examples have in common (e.g. all observed apples are small, round and red). The corresponding symbol ("apple") then becomes a "pointer" to the newly acquired concept `{small, round, red}` - a way to access and manipulate the concept without having to know its exact representational form. Our approach does not make any assumptions about how these symbols are encoded, which also allows SCAN to learn multiple referents to the same concept, i.e. synonyms.

Once a concept is acquired, it should be possible to use it for bi-directional inference: the model should be able to generate diverse visual samples that correspond to a particular concept (*sym2img*) and vice versa (*img2sym*). Since the projection from the space of visual primitives to the space of concepts (img2sym, red dash arrow in Fig. 1) involves abstraction and hence a loss of information, one then needs to add compatible information back in when moving from the space of concepts to that of visual primitives (sym2img, blue dot arrow in Fig. 1). In our setup, concepts are defined in terms of a set of relevant visual primitives (e.g. colour, shape and size for "apple"). This leaves a set of irrelevant visual attributes (e.g. lighting, position, background) to be "filled in". We do so by defaulting them to their respective priors, which ensures high diversity of samples (in both image or symbol space) for each concept during img2sym and sym2img inferences.

The structured nature of learnt concepts acquired by SCAN allows for sample efficient learning of logical recombination operators: AND (corresponding to a set union of relevant primitives), IN COMMON (corresponding to set intersection) and IGNORE (corresponding to set difference), by pairing a small number of valid visual examples of recombined concepts with the respective operator names. Once the meaning of the operators has been successfully learned, SCAN can exploit the compositionality of the acquired concepts, and traverse previously unexplored parts of the implicit underlying concept hierarchy by manipulating and recombining existing concepts in novel ways. For example, a new node corresponding to the concept `{blue, small}` can be reached through the following instructions: "blue" AND "small" (going down the hierarchy from more general to more

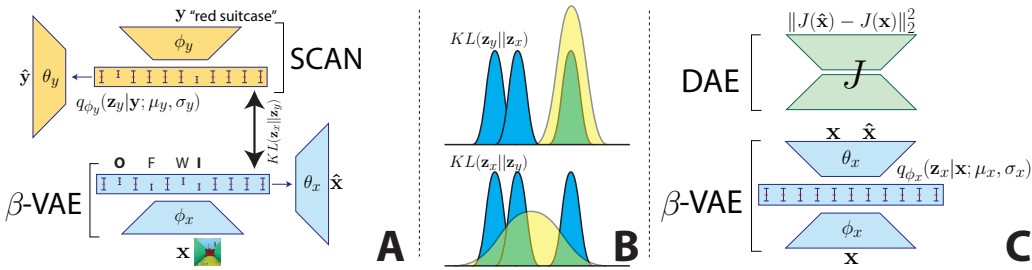

Figure 2: **A**: SCAN model architecture. The capital letters correspond to four disentangled visual primitives: object identity ($I$), object colour ($O$), floor colour ($F$) and wall colour ($W$). **B**: Mode coverage of the extra KL term of the SCAN loss function. Forward KL divergence $D_{KL}(\mathbf{z}_x \parallel \mathbf{z}_y)$ allows SCAN to learn abstractions (wide yellow distribution $\mathbf{z}_y$) over the visual primitives that are irrelevant to the meaning of a concept (blue modes corresponds to the inferred values of $\mathbf{z}_x$ for different visual examples matching symbol $\mathbf{y}$). **C**: $\beta$-VAE$_{DAE}$ model architecture.

specific), "blueberry" IN COMMON "bluebell" (going up the hierarchy from more specific to more general) or "blueberry" IGNORE "round" (also going up the hierarchy).

To summarise, our paper 1) presents SCAN, a neural network model capable of learning compositional and hierarchical representations of visual concepts; 2) demonstrates that SCAN can be successfully trained with very little supervised data; 3) shows that after training, SCAN can perform multimodal (visual and symbolic) bi-directional inference and generation with high accuracy and diversity, outperforming all baselines; 4) shows that the addition of logical recombination operations allows SCAN to break out of its limited training data distribution and reach new nodes within the implicit hierarchy of concepts.

## 2 RELATED WORK

To the best of our knowledge no framework currently exists that is directly equivalent to SCAN. Past relevant literature can broadly be split into three categories: 1) Bayesian models that try to mimic fast human concept learning (Tennenbaum, 1999; Lake et al., 2015); 2) conditional generative models that aim to generate faithful images conditioned on a list of attributes or other labels (Reed et al., 2016b;a; Kingma et al., 2014; Yan et al., 2016; Sohn et al., 2015; Pandey & Dukkipati, 2017) ; and 3) multimodal generative models that aim to embed visual and symbolic inputs in a joint latent space in order to be able to run bi-directional inferences (Vedantam et al., 2017; Suzuki et al., 2017; Pu et al., 2016; Wang et al., 2016; Srivastava & Salakhutdinov, 2014). Bayesian models by Tennenbaum (1999) and Lake et al. (2015) can learn from few examples, but, unlike SCAN, they are not fully grounded in visual data. Conditional and joint multimodal models are fully grounded in visual data, however, unlike SCAN, they require a large number of image-symbol pairs for training. An exception to this is the model by Srivastava & Salakhutdinov (2014), which, however, cannot generate images, instead relying on feature-guided nearest-neighbour lookup within existing data, and also requires slow MCMC sampling. Multimodal generative models are capable of bi-directional inference, however they tend to learn a flat unstructured latent space unlike the hierarchical compositional latent space of SCAN. Hence these baselines underperform SCAN in terms of sample diversity and the ability to break out of their training data distribution through symbolically instructed logical operations.

## 3 FORMALISING CONCEPTS

In Sec. 1 we informally proposed that concepts are abstractions over visual representational primitives. Hence, in order to formally define concepts we first define the visual representations used to ground them. These are defined as tuples of the form $(Z_1, ..., Z_K)$, where $\{1, ..., K\}$ is the set of indices of the independent latent factors sufficient to generate the visual input $\mathbf{x}$, and $Z_k$ is a random variable. The set $\mathbb{R}^K$ of all such tuples is a K-dimensional visual representation space.

We define a concept $C_i$ in such a K-dimensional representation space as a set of assignments of probability distributions to the random variables $Z_k$, with the following form:

$$C_i = \{(k, p_k^i(Z_k)) \mid k \in S_i\} \tag{1}$$

where $S_i \subseteq \{1, ..., K\}$ is the set of visual latent primitives that are relevant to concept $C_i$ and $p_k^i(Z_k)$ is a probability distribution specified for the visual latent factor represented by the random variable $Z_k$. Since $S_i$ are subsets of $\{1, ..., K\}$, concepts are abstractions over the K-dimensional visual representation space. To generate a visual sample corresponding to a concept $C_i$, it is necessary to fill in details for latents that got abstracted away during concept learning. This corresponds to the probability distributions $\{p_k(Z_k) | k \in \overline{S_i}\}$, where $\overline{S_i} = \{1, ..., K\} \setminus S_i$ is the set of visual latent primitives that are irrelevant to the concept $C_i$. In SCAN we set these to the unit Gaussian prior: $p_k(Z_k) = \mathcal{N}(0, 1), \ \forall k \in \overline{S_i}$.

In order to improve readability, we will use a simplified notation for concepts throughout the rest of the paper. For example, $\{ (size, \ p(Z_{size} = \text{small})), \ (colour, \ p(Z_{colour} = \text{blue})) \}$ will become either {small, blue} or {small, blue, _, _}, depending on whether we signify the irrelevant primitives $k \in \overline{S_i}$ as placeholder symbols "_". Note that unlike the formal notation, the ordering of attributes within the simplified notation is fixed and meaningful.

Since we define concepts as sets, we can also define binary relations and operators on these sets. If $C_1$ and $C_2$ are concepts, and $C_1 \subset C_2$, we say that $C_1$ is *superordinate* to $C_2$, and $C_2$ is *subordinate* to $C_1$. Two concepts $C_1$ and $C_2$ are *orthogonal* if $S_1 \cap S_2 = \varnothing$. The *conjunction* of two orthogonal concepts $C_1$ and $C_2$ is the concept $C_1 \cup C_2$ (e.g. {small, _, _} AND {_, round, _} = {small, round, _}). The *overlap* of two non-orthogonal concepts $C_1$ and $C_2$ is the concept $C_1 \cap C_2$ (e.g. {small, round, _} IN COMMON {_, round, red} = {_, round, _}). The *difference* between two concepts $C_1$ and $C_2$, where $C_1 \subset C_2$ is the concept $C_2 \setminus C_1$ (e.g. {small, round, _} IGNORE {_, round, _} = {small, _, _}). These operators allow for a traversal over a broader set of concepts within the implicit hierarchy given knowledge of a limited training subset of concepts.

## 4 MODEL ARCHITECTURE

**Learning visual representational primitives** The discovery of the generative structure of the visual world is the goal of disentangled factor learning research (Bengio et al., 2013). In this work we build SCAN on top of $\beta$-VAE, a state of the art model for unsupervised visual disentangled factor learning. $\beta$-VAE is a modification of the variational autoencoder (VAE) framework (Kingma & Welling, 2014; Rezende et al., 2014) that introduces an adjustable hyperparameter $\beta$ to the original VAE objective:

$$\mathcal{L}_x(\theta, \phi; \mathbf{x}, \mathbf{z}_x, \beta) = \mathbb{E}_{q_\phi(\mathbf{z}_x|\mathbf{x})}[\log p_\theta(\mathbf{x}|\mathbf{z}_x)] - \beta \, D_{KL}\big(q_\phi(\mathbf{z}_x|\mathbf{x}) \parallel p(\mathbf{z}_x)\big) \tag{2}$$

where $\phi, \theta$ parametrise the distributions of the encoder and the decoder respectively. Well chosen values of $\beta$ (usually $\beta > 1$) result in more disentangled latent representations $\mathbf{z}_x$ by setting the right balance between reconstruction accuracy, latent channel capacity and independence constraints to encourage disentangling. For some datasets, however, this balance is tipped too far away from reconstruction accuracy. In these scenarios, disentangled latent representations $\mathbf{z}_x$ may be learnt at the cost of losing crucial information about the scene, particularly if that information takes up a small proportion of the observations $\mathbf{x}$ in pixel space. Hence, we adopt the solution used in Higgins et al. (2017b) that replaces the pixel log-likelihood term in Eq. 2 with an L2 loss in the high-level feature space of a denoising autoencoder (DAE) (Vincent et al., 2010) trained on the same data (see Fig. 2C for model architecture). The resulting $\beta$-VAE$_{DAE}$ architecture optimises the following objective function:

$$\mathcal{L}_x(\theta, \phi; \mathbf{x}, \mathbf{z}_x, \beta) = \mathbb{E}_{q_\phi(\mathbf{z}_x|\mathbf{x})} \|J(\hat{\mathbf{x}}) - J(\mathbf{x})\|_2^2 - \beta \, D_{KL}\big(q_\phi(\mathbf{z}_x|\mathbf{x}) \parallel p(\mathbf{z}_x)\big) \tag{3}$$

where $\hat{\mathbf{x}} \sim p_\theta(\mathbf{x}|\mathbf{z}_x)$ and $J : \mathbb{R}^{W \times H \times C} \to \mathbb{R}^N$ is the function that maps images from pixel space with dimensionality Width $\times$ Height $\times$ Channels to a high-level feature space with dimensionality $N$ given by a stack of DAE layers up to a certain layer depth (a hyperparameter). Note that this adjustment means that we are no longer optimising the variational lower bound, and $\beta$-VAE$_{DAE}$ with $\beta = 1$ loses its equivalence to the original VAE framework.

**Learning visual concepts**    This section describes how our proposed SCAN framework (Fig. 2A) exploits the particular parametrisation of the visual building blocks acquired by $\beta$-VAE[1] to learn an implicit hierarchy of visual concepts formalised in Sec. 3. SCAN is based on a modified VAE framework. In order to encourage the model to learn visually grounded abstractions, we initialise the space of concepts (the latent space $\mathbf{z}_y$ of SCAN) to be structurally identical to the space of visual primitives (the latent space $\mathbf{z}_x$ of $\beta$-VAE). Both spaces are parametrised as multivariate Gaussian distributions with diagonal covariance matrices, and $\dim(\mathbf{z}_y) = \dim(\mathbf{z}_x) = K$. The grounding is performed by aiming to minimise the KL divergence between the two distributions.

The abstraction step corresponds to setting SCAN latents $z_y^k$ corresponding to the relevant factors to narrow distributions, while defaulting those corresponding to the irrelevant factors to the wider unit Gaussian prior. This is done by minimising the forward KL divergence $D_{KL}\big(q(\mathbf{z}_x) \parallel q(\mathbf{z}_y)\big)$, rather than the mode picking reverse KL divergence $D_{KL}\big(q(\mathbf{z}_y) \parallel q(\mathbf{z}_x)\big)$. Fig. 2B demonstrates the differences. Each blue mode corresponds to an inferred visual latent distribution $q(z_x^k|x_i)$ given an image $x_i$. The yellow distribution corresponds to the learnt conceptual latent distribution $q(z_y^k)$. When presented with visual examples that have high variability for a particular generative factor, e.g. various lighting conditions when viewing examples of apples, the forward KL allows SCAN to learn a broad distribution for the corresponding conceptual latent $q(z_y^k)$ that is close to the prior $p(z_y^k) = \mathcal{N}(0,1)$. Hence, SCAN is trained by minimising:

$$\mathcal{L}_y(\theta_y, \phi_y; \mathbf{y}, \mathbf{x}, \mathbf{z}_y, \beta, \lambda) = \mathbb{E}_{q_{\phi_y}(\mathbf{z}_y|\mathbf{y})}[\log p_{\theta_y}(\mathbf{y}|\mathbf{z}_y)] - \beta\, D_{KL}\big(q_{\phi_y}(\mathbf{z}_y|\mathbf{y}) \parallel p(\mathbf{z}_y)\big)$$
$$- \lambda\, D_{KL}\big(q_{\phi_x}(\mathbf{z}_x|\mathbf{x}) \parallel q_{\phi_y}(\mathbf{z}_y|\mathbf{y})\big) \tag{4}$$

where $\mathbf{y}$ is symbol inputs, $\mathbf{z}_y$ is the latent space of concepts, $\mathbf{z}_x$ is the latent space of the pre-trained $\beta$-VAE containing the visual primitives which ground the abstract concepts $\mathbf{z}_y$, and $\mathbf{x}$ are example images that correspond to the concepts $\mathbf{z}_y$ activated by symbols $\mathbf{y}$. It is important to up-weight the forward KL term relative to the other terms in the cost function (e.g. $\lambda = 1$, $\beta = 10$).

The SCAN architecture does not make any assumptions on the nature of the symbols $\mathbf{y}$. In this paper we use a commonly used k-hot encoding (Vedantam et al., 2017; Suzuki et al., 2017), where each concept is described in terms of the $k \leq K$ visual attributes it refers to (e.g. an apple could be referred to by a 3-hot symbol "round, small, red"). In principle, other possible encoding schemes for $\mathbf{y}$ can also be used, including word embeddings (Mikolov et al., 2013), or even entirely random vectors. We leave the empirical demonstration of this to future work.

Once trained, SCAN allows for bi-directional inference and generation (img2sym and sym2img). In order to generate visual samples that correspond to a particular concept (sym2img), we infer the concept $\mathbf{z}_y$ by presenting an appropriate symbol $\mathbf{y}$ to the inference network of SCAN. One can then sample from the inferred concept $q_{\phi_y}(\mathbf{z}_y|\mathbf{y})$ and use the generative part of $\beta$-VAE to visualise the corresponding image samples $p_{\theta_x}(\mathbf{x}|\mathbf{z}_y)$. SCAN can also be used to infer a description of an image in terms of the different learnt concepts via their respective symbols. To do so, an image $\mathbf{x}$ is presented to the inference network of the $\beta$-VAE to obtain its description in terms of the visual primitives $\mathbf{z}_x$. One then uses the generative part of the SCAN to sample descriptions $p_{\theta_y}(\mathbf{y}|\mathbf{z}_x)$ in terms of symbols that correspond to the previously inferred visual building blocks $q_{\phi_x}(\mathbf{z}_x|\mathbf{x})$.

**Learning concept recombination operators**    The compositional and hierarchical structure of the concept latent space $\mathbf{z}_y$ learnt by SCAN can be exploited to break away from the training data distribution and imagine new concepts. This can be done by using logical concept manipulation operators AND, IN COMMON and IGNORE formally defined in Sec. 3. These operators are implemented within a conditional convolutional module parametrised by $\psi$ (Fig. 3A) that accepts two multivariate Gaussian distributions $\mathbf{z}_{y_1}$ and $\mathbf{z}_{y_2}$ corresponding to the two concepts that are to be recombined, and a conditioning vector $\mathbf{r}$ specifying the recombination operator. The input distributions $\mathbf{z}_{y_1}$ and $\mathbf{z}_{y_2}$ are inferred from the two corresponding input symbols $\mathbf{y}_1$ and $\mathbf{y}_2$, respectively, using a pre-trained SCAN. The convolutional module strides over the parameters of each matching component $z_{y_1}^k$ and $z_{y_2}^k$ one at a time and outputs the corresponding parametrised component $z_r^k$ of a recombined multivariate Gaussian distribution $\mathbf{z}_r$ with a diagonal covariance matrix.[2] We used 1-hot encoding

---

[1] For the rest of the paper we use the term $\beta$-VAE to refer to $\beta$-VAE$_{DAE}$.

[2] We also tried a closed form implementation of recombination operators (weighted sum or mean of the corresponding Gaussian components $z_{y_1}^k$ and $z_{y_2}^k$). We found that the learnt recombination operators worked

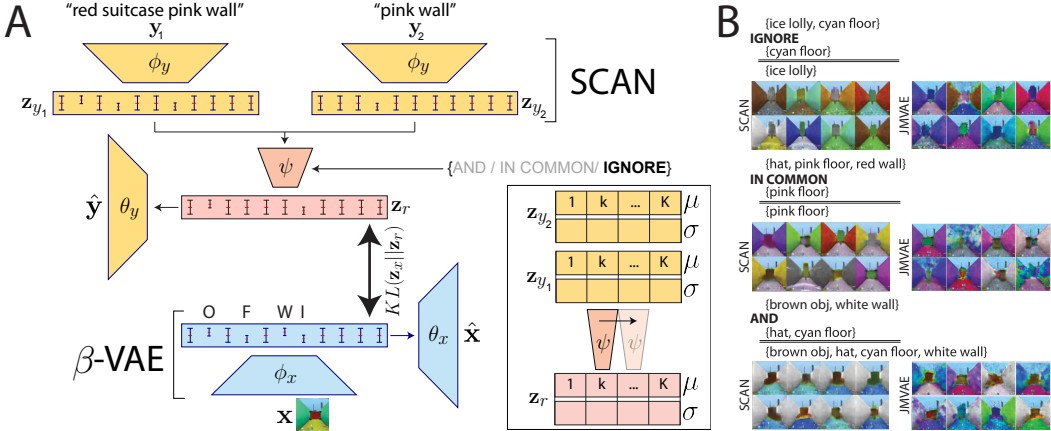

Figure 3: **A**: Learning AND, IN COMMON or IGNORE recombination operators with a SCAN model architecture. Inset demonstrates the convolutional recombination operator that takes in $\{\mu_{y_1}^k, \sigma_{y_1}^k; \mu_{y_2}^k, \sigma_{y_2}^k\}$ and outputs $\{\mu_r^k, \sigma_r^k\}$. The capital letters correspond to four disentangled visual primitives: object identity ($I$), object colour ($O$), floor colour ($F$) and wall colour ($W$). **B**: Visual samples produced by SCAN and JMVAE when instructed with a novel concept recombination. SCAN samples consistently match the expected ground truth recombined concept, while maintaining high variability in the irrelevant visual primitives. JMVAE samples lack accuracy. Recombination instructions are used to imagine concepts that have never been seen during model training. **Top**: samples for IGNORE; **Middle**: samples for IN COMMON; **Bottom**: samples for AND.

for the conditioning vector $\mathbf{r}$, where $[\,1\ 0\ 0\,]$, $[\,0\ 1\ 0\,]$ and $[\,0\ 0\ 1\,]$ stood for AND, IN COMMON and IGNORE respectively. The conditioning was implemented as a tensor product that takes in $\mathbf{z}_{y_1}$ and $\mathbf{z}_{y_2}$ and outputs $\mathbf{z}_r$, where $\mathbf{r}$ effectively selects the appropriate trainable transformation matrix parametrised by $\psi$. The conditional convolutional module is trained through the same visual grounding process as SCAN– each recombination instruction is paired with a small number of appropriate example images (e.g. "blue,suitcase" IGNORE "suitcase" might be paired with various example images containing a blue object). The recombination module is trained by minimising:

$$\mathcal{L}_r(\psi; \mathbf{z}_x, \mathbf{z}_r) = D_{KL}\big[q_{\phi_x}(\mathbf{z}_x|\mathbf{x}_i) \,\big|\big|\, q_\psi\big(\mathbf{z}_r \mid q_{\phi_y}(\mathbf{z}_{y_1}|\mathbf{y}_1), q_{\phi_y}(\mathbf{z}_{y_2}|\mathbf{y}_2), \mathbf{r}\big)\big] \tag{5}$$

where $q_{\phi_x}(\mathbf{z}_x|\mathbf{x}_i)$ is the inferred latent distribution of the $\beta$-VAE given a seed image $\mathbf{x}_i$ that matches the specified symbolic description. The resulting $\mathbf{z}_r$ lives in the same space as $\mathbf{z}_y$ and corresponds to a node within the implicit hierarchy of visual concepts. Hence, all the properties of concepts $\mathbf{z}_y$ discussed in the previous section also hold for $\mathbf{z}_r$.

## 5 EXPERIMENTS

### 5.1 DEEPMIND LAB EXPERIMENTS

**Environment** We evaluate the performance of SCAN on a dataset of visual frames and corresponding symbolic descriptions collected within the DeepMind Lab environment (Beattie et al., 2016). DeepMind Lab was chosen, because it gave us good control of the ground truth generative process. The visual frames were collected from a static viewpoint situated in a room containing a single object. The generative process was specified by four factors of variation: wall colour, floor colour, object colour with 16 possible values each, and object identity with 3 possible values: hat, ice lolly and suitcase. Other factors of variation were also added to the dataset by the DeepMind Lab engine, such as the spawn animation, horizontal camera rotation and the rotation of objects around the vertical axis. We split the dataset into two subsets. One was used for training the models, while the other one contained a held out set of 300 four-gram concepts that were never seen during training, either visually or symbolically. We used the held out set to evaluate the model's ability to imagine new concepts.

better, achieving 0.79 vs 0.54 accuracy (higher is better) and 1.05 vs 2.03 diversity (lower is better) scores compared to the closed form implementations. See Sec. 5.1 for the description of the accuracy and diversity metrics.

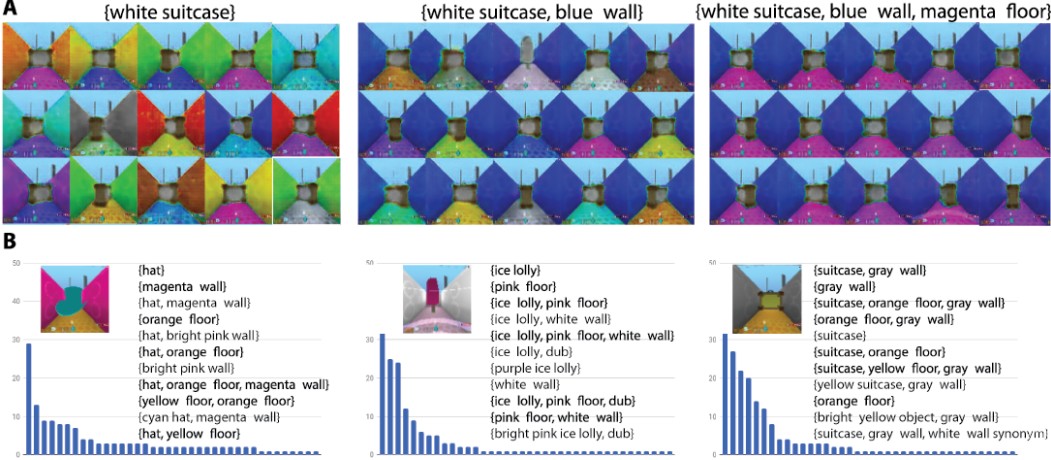

Figure 4: **A**: sym2img inferences with "white suitcase", "white suitcase, blue wall", and "white suitcase, blue wall, magenta floor" as input. The latter one points to a concept that the model has never seen during training, either visually or symbolically. All samples are consistently accurate, while showing good diversity in terms of the irrelevant visual attributes. **B**: when presented with an image, SCAN is able to describe it in terms of all concepts it has learnt, including synonyms (e.g. "dub", which corresponds to {ice lolly, white wall}). The histograms show the distributions of unique concepts the model used to describe each image, most probable of which are printed in descending order next to the corresponding image. The few confusions SCAN makes are intuitive to humans too (e.g. confusing orange and yellow colours).

**Learning grounded concepts** In this section we demonstrate that SCAN is capable of learning the meaning of new concepts from very few image-symbol pairs. We evaluate the model's concept understanding through qualitative analysis of sym2img and img2sym samples. First we pre-trained a $\beta$-VAE to learn a disentangled representation of the DeepMind Lab dataset (see Sec. A.3 in Supplemenrary Materials for details). Then we trained SCAN on a random subset of 133 out of 18,883 possible concepts sampled from all levels of the implicit hierarchy (these concepts specify between one and four visual primitives, and are associated with 1- to 4-hot symbols respectively). The set of symbols also included a number of 1-hot synonyms (e.g. a blue wall may be described by symbols "blue wall", "bright blue wall" or "blue wall synonym"). Each concept was associated with ten visual examples during training.

Fig. 4A shows samples drawn from SCAN when asked to imagine a bigram concept {white, suitcase}, a trigram concept {white, suitcase, blue wall}, or a four-gram {white, suitcase, blue wall, magenta floor}. Note that the latter is a concept drawn from the held-out test set that neither $\beta$-VAE nor SCAN have ever seen during training, and the first two concepts are novel to SCAN, but have been experienced by $\beta$-VAE. It is evident that the model demonstrates a good understanding of all three concepts, producing visual samples that match the meaning of the concept, and showing good variability over the irrelevant factors. Confusions do sometimes arise due to the sampling process (e.g. one of the suitcase samples is actually an ice lolly). Fig. 4B demonstrates that the same model can also correctly describe an image. The labels are mostly consistent with the image and display good diversity (SCAN is able to describe the same image using different symbols including synonyms). The few confusions that SCAN does make are between concepts that are easily confusable for human too (e.g. red, orange and yellow colours).

**Evolution of concept understanding** In this section we take a closer look inside SCAN as it learns a new concept. In Sec. 3 we suggested that concepts should be grounded in terms of specified factors (the corresponding latent units $z_y^k \; \forall k \in S$ should have low inferred standard deviations $\sigma_y^k$), while the unspecified visual primitives should be sampled from the unit Gaussian prior (the corresponding latent units $z_y^k \; \forall k \in \overline{S}$ show have $\sigma_y^k \approx 1$). We visualise this process by teaching SCAN the meaning of the concept {cyan wall} using a curriculum of fifteen progressively more diverse visual examples (see Fig. 5, bottom row). After training SCAN on each set of five visual examples, we test the model's understanding of the concept through sym2img sampling using the symbol "cyan wall" (Fig. 5, top four rows). We also plot the average inferred specificity of all 32 latent units $z_y^k$ during training

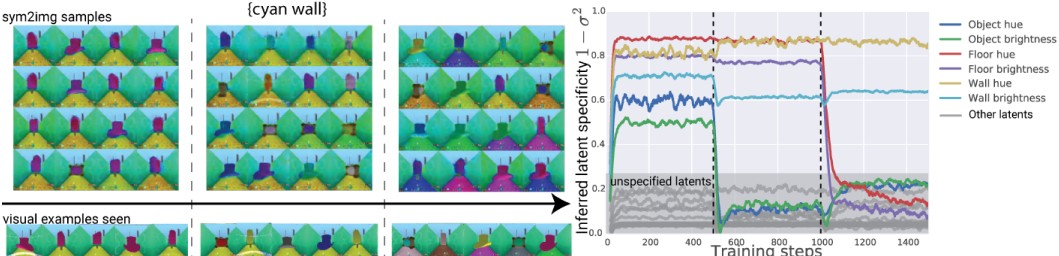

Figure 5: Evolution of understanding of the meaning of concept {cyan wall} as SCAN is exposed to progressively more diverse visual examples. **Left**: top row contains three sets of visual samples (sym2img) generated by SCAN after seeing each set of five visual examples presented in the bottom row. **Right**: average inferred specificity of concept latents $z_y^k$ during training. Vertical dashed lines correspond to the vertical dashed lines in the left plot and indicate a switch to the next set of five more diverse visual examples. 6/32 latents $z_y^k$ and labelled according to their corresponding visual primitives in $\mathbf{z}_x$.

(Fig. 5, right). It can be seen that the number of specified latents $z_y^k$ drops from six, over four, to two as the diversity of visual examples seen by SCAN increases. The remaining two highly specified latents $z_y^k$ correctly correspond to the visual primitives $\mathbf{z}_x$ representing wall hue and brightness.

**Quantitative comparison to baselines**   In this section we quantitatively compare the accuracy and diversity of the sym2img samples produced by the SCAN to those of the baselines – a SCAN like architecture trained with a reverse KL used for grounding conceptual representations in vision (SCAN$_R$), another modification of SCAN that tries to ground conceptual representations in unstructured (entangled) visual representations (SCAN$_U$, with various levels of visual entanglement), and two of the latest multimodal joint density models, the JMVAE (Suzuki et al., 2017) and the triple ELBO (TrELBO) (Vedantam et al., 2017). The two metrics, accuracy and diversity, measure different aspects of the models' performance. High accuracy means that the models understand the meaning of a symbol (e.g. samples of a "blue suitcase" should contain blue suitcases). High diversity means that the models were able to learn an abstraction. It quantifies the variety of samples in terms of the unspecified visual attributes (e.g. samples of blue suitcases should include a high diversity of wall colours and floor colours). There is a correlation between the two metrics, since samples with low accuracy often result in higher diversity scores.

We use a pre-trained classifier achieving $99\%$ average accuracy over all data generative factors to evaluate the accuracy of img2sym samples. Since some colours in the dataset are hard to differentiate even to humans (e.g. yellow and orange), we use top-3 accuracy for colour related factors. We evaluate the diversity of visual samples by estimating the KL divergence of the inferred factor distribution with the flat prior: $D_{KL}\big(u(\mathbf{y}_i) \parallel p(\mathbf{y}_i)\big)$, where $p(\mathbf{y}_i)$ is the joint distribution over the factors irrelevant to the $i$th concept $i \in \overline{S_i}$ (inferred by the classifier) and $u(\mathbf{y}_i)$ is the equivalent flat distribution (i.e., with each factor value having equal probability). See Sec. A.1 in Supplementary Materials for more details.

All models were trained on a random subset of 133 out of 18,883 possible concepts sampled from all levels of the implicit hierarchy with ten visual examples each. The accuracy and diversity metrics were calculated on two sets of sym2img samples: 1) *train*, corresponding to the 133 symbols used to train the models; and 2) *test (symbols)*, corresponding to a held out set of 50 symbols. Tbl. 1 demonstrates that SCAN outperforms all baselines in terms of both metrics. SCAN$_R$ learns very accurate representations, however it overfits to a single mode of each of the irrelevant visual factors and hence lacks diversity. SCAN$_U$ experiments show that as the level of disentanglement within the visual representation is increased (the higher the $\beta$, the more disentangled the representation), the accuracy and the diversity of the sym2img samples also get better. Note that baselines with poor sample accuracy inadvertently have good diversity scores because samples that are hard to classify produce a relatively flat classifier distribution $p(\mathbf{y}_i)$ close to the uniform prior $u(\mathbf{y}_i)$. TrELBO learns an entangled and unstructured conceptual representation that produces accurate yet stereotypical sym2img samples that lack diversity. Finally, JMVAE is a model that comes the closest to SCAN

| MODEL | ACCURACY | | | DIVERSITY | | |
|---|---|---|---|---|---|---|
| | TRAIN | TEST (SYMBOLS) | TEST (OPERATORS) | TRAIN | TEST (SYMBOLS) | TEST (OPERATORS) |
| TRELBO | 0.81 | 0.69 | 0.37 | 9.41 | 6.86 | **0.63** |
| JMVAE | 0.75 | 0.68 | 0.61 | 4.32 | 2.87 | 0.86 |
| SCAN$_R$ | **0.86** | **0.81** | 0.67 | 13.17 | 9.2 | 9.94 |
| SCAN$_U$ ($\beta = 0.1$) | 0.27 | 0.26 | 0.25 | 5.51 | 1.23 | 1.66 |
| SCAN$_U$ ($\beta = 1$) | 0.58 | 0.36 | 0.33 | 2.07 | 1.22 | 1.34 |
| SCAN$_U$ ($\beta = 20$) | 0.65 | 0.42 | 0.32 | **1.41** | 3.98 | 4.57 |
| **SCAN** ($\beta = 53$) | 0.82 | 0.79 | **0.79** | 1.46 | **1.08** | 1.05 |

Table 1: Quantitative results comparing the accuracy and diversity of visual samples produced through sym2img inference by SCAN and three baselines: a SCAN with unstructured vision (SCAN$_U$, lower $\beta$ means more visual entanglement), a SCAN with a reverse grounding KL term for both the model itself and its recombination operator (SCAN$_R$) and two recent joint multimodal embedding models, JMVAE and TrELBO. Higher accuracy and lower diversity indicate better performance. Test values can be computed either by directly feeding the ground truth symbols (*test symbols*), or by applying trained recombination operators to make the model recombine in the latent space (*test operators*).

in terms of performance. It manages to exploit the structure of the symbolic inputs to learn a representation of the joint posterior that is almost as disentangled as that of SCAN. Similarly to SCAN, it also uses a forward KL term to match unimodal posteriors to the joint posterior. Hence, given that there is enough supervision within the symbols to help JMVAE learn a disentangled joint posterior, it should become equivalent to SCAN, whereby the joint $q(\mathbf{z}|\mathbf{x}, \mathbf{y})$ and unimodal $q(\mathbf{z}|\mathbf{y})$ posteriors of JMVAE become equivalent to the visual $q(\mathbf{z}_x|\mathbf{x})$ and symbolic $q(\mathbf{z}_y|\mathbf{y})$ posteriors of SCAN respectively. Yet in practice we found that JMVAE training is much more sensitive to various architectural and hyperparameter choices compared to SCAN, which often results in mode collapse leading to the reasonable accuracy yet poor diversity of the JMVAE sym2img samples. See Sec. A.3 for more details of the baselines' performance. Finally, SCAN is the only model that was able to exploit the k-hot structure of the symbols and the compositional nature of its representations to generalise well to the test set (*test symbols* results), while all of the other baselines lost a lot of their sample accuracy.

**Learning recombination operators** In Sec. 4 we suggested a way to traverse the implicit hierarchy of concepts towards novel nodes without any knowledge of how to point to these nodes through a symbolic reference. We suggested doing so by instructing a recombination of known concepts in the latent space. To test this, we trained a recombination module using 10 recombination instructions per each of the three operators, with 20 visual examples each. Tbl. 1 (*test operators*) demonstrates that we were able to reach the nodes corresponding to the 50 novel test concepts using such a pre-trained recombination operator module. This, however, only worked for SCAN, since the successful training of the recombination module relies on a structured latent space that all the other baselines lacked. SCAN with the recombination module preserved the accuracy and the diversity of samples, as shown quantitatively in Tbl. 1 and qualitatively in Fig. 3B. JMVAE, the closest baseline to SCAN in terms of the recombination module performance, produced samples with low accuracy (the drop in accuracy resulted in an increase in the diversity score). It is interesting to note that the recombination operator training relies on the same kind of visual grounding as SCAN, hence it can often improve the diversity of the original model.

## 5.2 CELEBA EXPERIMENTS

We ran additional experiments on a more realistic dataset CelebA (Liu et al., 2015) after performing minimal dataset pre-processing of cropping the frames to 64x64. Unlike other approaches (Vedantam et al., 2017; Perarnau et al., 2016) which only use 18 best attributes for training their models, we used all 40 attributes. Many of these 40 attributes are not useful, since they are either: 1) subjective (e.g. "attractiveness"); 2) refer to parts of the image that have been cropped out (e.g. "wearing necktie"); 3) refer to visual features that have not been discovered by $\beta$-VAE (e.g. "sideburns", see Higgins et al. (2017a) for a discussion of the types of factors that $\beta$-VAE tends to learn on this dataset); 4) are confusing due to mislabelling (e.g. "bald female" as reported by Vedantam et al. (2017)). Hence, our experiments test the robustness of SCAN to learning concepts in an adversarial setting, where the model is taught concepts that do not necessarily relate well to their corresponding visual examples. For these experiments we used the controlled capacity schedule (Burgess et al., 2017) for $\beta$-VAE training to increase the quality of the generative process of the model.

We found that SCAN trained on CelebA was able to outperform its baselines of JMVAE and TrELBO. First, we checked which of the 40 attributes SCAN was able to understand after training. To do so,

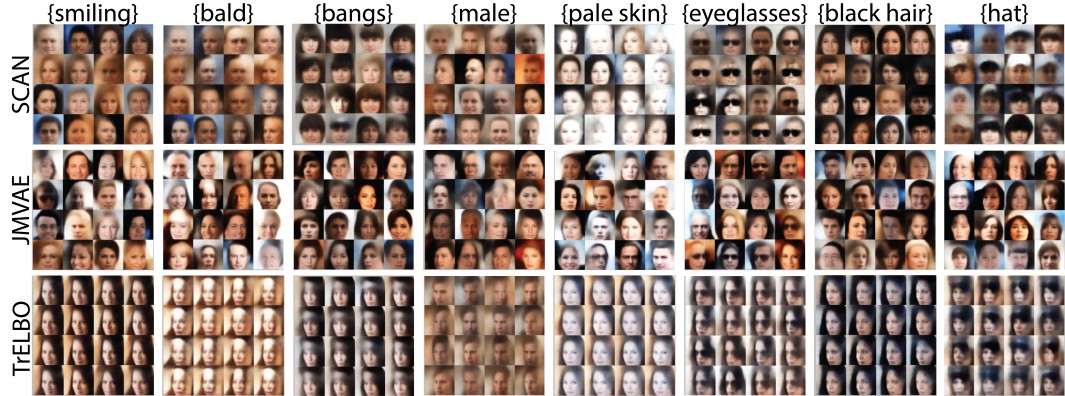

Figure 6: Comparison of sym2img samples of SCAN, JMVAE and TrELBO trained on CelebA. See Fig. 19 in Supplementary Materials for larger samples.

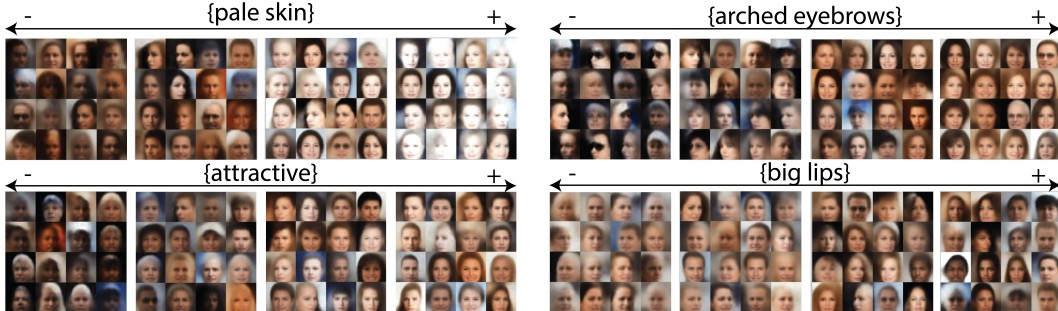

Figure 7: Example sym2img samples of SCAN trained on CelebA. We run inference using four different values for each attribute. We found that the model was more sensitive to changes in values in the positive rather than negative direction, hence we use the following values: $\{-6, -3, 1, 2\}$. See Fig. 20 in Supplementary Materials for larger samples.

we inferred $q(\mathbf{z}_y|\mathbf{y}_i)$ for all $\mathbf{y}_i \in \mathcal{R}^{40}$, where $\mathbf{y}_i$ is a 1-hot encoding of the ith attribute. We then approximated the number of *specified* latents for each posterior $q(\mathbf{z}_y|\mathbf{y}_i)$. If an attribute $i$ did not correspond to anything meaningful in the corresponding visual examples seen during training, it would have no *specified* latents and $D_{KL}(q(\mathbf{z}_y|\mathbf{y}_i)||p(\mathbf{z}_y)) \approx 0$. We found that SCAN did indeed learn the meaning of a large number of attributes. Fig. 6 shows *sym2img* samples for some of them compared to the equivalent samples for the baseline models: JMVAE and TrELBO. It can be seen that SCAN samples tend to be more faithful than those of JMVAE, and both models produce much better diversity of samples than TrELBO.

A notable difference between SCAN and the two baselines is that despite being trained on *binary* k-hot attribute vectors (where k varies for each sample), SCAN learnt meaningful directions of *continuous* variability in its conceptual latent space $\mathbf{z}_y$. For example, if we vary the value of an individual symbolic attribute, we will get meaningful sym2img samples that range between extreme positive and extreme negative examples of that attribute (e.g. by changing the values of the "pale skin" symbol $\mathbf{y}$, we can generate samples with various skin tones as shown in Fig. 7). This is in contrast to JMVAE and TrELBO, which only produce meaningful sym2img samples if the value of the attribute is set to 1 (attribute is present) or 0 (attribute is not enforced). This means that unlike SCAN, it is impossible to enforce JMVAE or TrELBO to generate samples with darker skin colours despite the models knowing the meaning of the "pale skin" attribute.

Note that sometimes SCAN picks up implicit biases in the dataset. For example, after training SCAN interprets "attractive" as a term that refers to young white females and less so to males, especially if these males are also older and have darker skin tones (Fig. 7). Similarly, SCAN learns to use the term "big lips" to describe younger ethnic individuals, and less so older white males; while "arched eyebrows" is deemed appropriate to use when describing young white females, but not when

describing people wearing sunglasses or hats, presumably because one cannot see how arched their eyebrows are.

## 6 CONCLUSION

This paper introduced a new approach to learning grounded visual concepts. We defined concepts as abstractions over independent (and often interpretable) visual primitives, where each concept is given by learned distributions over a set of relevant visual factors. We proposed that all other (irrelevant) visual factors should default to their prior in order to produce a diverse set of samples corresponding to a concept. We then proposed SCAN, a neural network implementation of such an approach, which was able to discover and learn an implicit hierarchy of abstract concepts from as few as five symbol-image pairs per concept and no assumptions on the nature of symbolic representations. SCAN was then capable of bi-directional inference, generating diverse and accurate image samples from symbolic instructions, and vice versa, qualitatively and quantitatively outperforming all baselines, including on a realistic CelebA dataset with noisy attribute labels. The structure of the learnt concepts allowed us to train an extension to SCAN that could perform logical recombination operators. We demonstrated how such operators could be used to traverse the implicit concept hierarchy, including imagining completely new concepts. Due to the sample efficiency and the limited number of assumptions in our approach, the representations learnt by SCAN should be immediately applicable within a large set of broader problem domains, including reinforcement learning, classification, control and planning.

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

# A   SUPPLEMENTARY INFORMATION

## A.1   MODEL DETAILS

$\beta$-**VAE**   We re-used the architecture and the training setup for $\beta$-VAE specified in Higgins et al. (2017b). In particular, we used L2 loss within a pre-trained denoising autoencoder (DAE) to calculate the reconstruction part of the $\beta$-VAE loss function. The DAE was trained with occlusion-style masking noise in the vein of Pathak et al. (2016). Concretely, two values were independently sampled from $U[0, W]$ and two from $U[0, H]$ where $W$ and $H$ were the width and height of the input frames. These four values determined the corners of the rectangular mask applied; all pixels that fell within the mask were set to zero.

The DAE architecture consisted of four convolutional layers, each with kernel size 4 and stride 2 in both the height and width dimensions. The number of filters learnt for each layer was $\{32, 32, 64, 64\}$ respectively. The bottleneck layer consisted of a fully connected layer of size 100 neurons. This was followed by four deconvolutional layers, again with kernel sizes 4, strides 2, and $\{64, 64, 32, 32\}$ filters. The padding algorithm used was 'SAME' in TensorFlow (Abadi et al., 2015). ELU non-linearities were used throughout. The optimiser used was Adam (Kingma & Ba, 2015) with a learning rate of 1e−3 and $\epsilon = 1$e−8. We pre-trained the DAE for 200,000 steps, using batch size of 100 before training $\beta$-VAE.

$\beta$-VAE architecture was the following. We used an encoder of four convolutional layers, each with kernel size 4, and stride 2 in the height and width dimensions. The number of filters learnt for each layer was $\{32, 32, 64, 64\}$ respectively. This was followed by a fully connected layer of size 256 neurons. The latent layer comprised 64 neurons parametrising 32 (marginally) independent Gaussian distributions. The decoder architecture was simply the reverse of the encoder, utilising deconvolutional layers. The decoder used was Bernoulli. The padding algorithm used was 'SAME' in TensorFlow. ReLU non-linearities were used throughout. The reconstruction error was taking in the last layer of the DAE (in the pixel space of DAE reconstructions) using L2 loss and before the non-linearity. The optimiser used was Adam with a learning rate of 1e−4 and $\epsilon = 1$e−8. We pre-trained $\beta$-VAE until convergence using batch size of 100. The disentangled $\beta$-VAE had $\beta = 53$, while the entangled $\beta$-VAE used within the SCAN$_U$ baseline had $\beta = 0.1$.

**SCAN**   The encoder and decoder of SCAN were simple single layer MLPs with 100 hidden units for DeepMind Lab experiments and a two layer MLP with 500 hidden units in each hidden layer for the CelebA experiments. We used ReLU non-linearities in both cases. The decoder was parametrised as a Bernoulli distribution over the output space of size 375. We set $\beta_y = 1$ for all experiments, and $\lambda = 10$. We trained the model using Adam optimiser with learning rate of 1e−4 and batch size 16.

**SCAN recombination operator**   The recombination operator was implemented as a convolutional operator with kernel size 1 and stride 1. The operator was parametrised as a 2 layer MLP with 30 and 15 hidden units per layer, and ReLU non-linearities. The optimizer is Adam with a learning rate of 1e−3 and batch size 16 was used. We trained the recombination operator for $50k$ steps.

**JMVAE**   The JMVAE was trained using the loss as described in (Suzuki et al., 2017):

$$
\begin{aligned}
\mathcal{L}_{JM}(\theta_x, \theta_y, \phi_x, \phi_y, \phi; \mathbf{x}, \mathbf{y}, \alpha) = {} & \mathbb{E}_{q_\phi(\mathbf{z}|\mathbf{x},\mathbf{y})}\left[\log p_{\theta_x}(\mathbf{x}|\mathbf{z})\right] \; + \; \mathbb{E}_{q_\phi(\mathbf{z}|\mathbf{x},\mathbf{y})}\left[\log p_{\theta_y}(\mathbf{y}|\mathbf{z})\right] \\
& - D_{KL}\big(q_\phi(\mathbf{z}|\mathbf{x},\mathbf{y}) \parallel p(\mathbf{z})\big) \\
& - \alpha\big[D_{KL}\big(q_\phi(\mathbf{z}|\mathbf{x},\mathbf{y}) \parallel q_{\phi_x}(\mathbf{z}|\mathbf{x})\big) \\
& \quad + D_{KL}\big(q_\phi(\mathbf{z}|\mathbf{x},\mathbf{y}) \parallel q_{\phi_y}(\mathbf{z}|\mathbf{y})\big)\big]
\end{aligned} \tag{6}
$$

Where $\alpha$ was a hyperparameter. We tried $\alpha$ values $\{0.01, 0.1, 1.0, 10.0\}$ as in the original paper and found that the best results were obtained with $\alpha = 1.0$. All results were reported with this value.

The architectural choices for JMVAE were made to match as closely as possible those made for SCAN. Thus the visual encoder $q_{\phi_x}$ consisted of four convolutional layers, each with kernel size 4 and stride 2 in both the height and width dimensions, with $\{32, 32, 64, 64\}$ filters learned at the respective layers. The convolutional stack was followed by a single fully connected layer with 256 hidden units. The encoder output the parametrisation for a a 32-dimensional diagonal Gaussian latent distribution. The symbol encoder $q_{\phi_y}$ consisted of a single layer MLP with 100 hidden units for the DeepMind Lab experiments or two layer MLP with 500 hidden units per layer for the CelebA experiments as in SCAN. The joint encoder $q_\phi$ consisted of the same convolutional stack as in the visual encoder to process the visual input, while the symbol input was passed through a two-layer MLP with 32 and 100 hidden units. These two embeddings were then concatenated and passed through a further two-layer MLP of 256 hidden units each, before outputting the 64 parameters of the diagonal Gaussian latents.

The visual decoder $p_{\theta_x}$ was simply the reverse of the visual encoder using transposed convolutions. Similarly, the symbol decoder $p_{\theta_x}$ was again a single layer MLP with 100 hidden units. The output distributions of both decoders were parameterised as Bernoullis. The model was trained using the Adam optimiser with a learning rate of $1e-4$ and a batch size of 16.

**Triple ELBO (TrELBO)**   The Triple ELBO (TrELBO) model was trained using the loss as described in (Vedantam et al., 2017):

$$
\begin{aligned}
\mathcal{L}_{trelbo}(\theta_x, \theta_y, \phi_x, \phi_y, \phi; \mathbf{x}, \mathbf{y}, \lambda_y^{yx}, \lambda_y^y) = \ & \mathbb{E}_{q_\phi(\mathbf{z}|\mathbf{x},\mathbf{y})} \left[\log p_{\theta_x}(\mathbf{x}|\mathbf{z})\right] \ + \ \mathbb{E}_{q_{\phi_x}(\mathbf{z}|\mathbf{x})} \left[\log p_{\theta_x}(\mathbf{x}|\mathbf{z})\right] \\
& + \lambda_y^{yx} \mathbb{E}_{q_\phi(\mathbf{z}|\mathbf{x},\mathbf{y})} \left[\log p_{\theta_y}(\mathbf{y}|\mathbf{z})\right] + \lambda_y^y \mathbb{E}_{q_{\phi_y}(\mathbf{z}|\mathbf{y})} \left[\log p_{\theta_y}(\mathbf{y}|\mathbf{z})\right] \\
& - D_{KL}\big(q_\phi(\mathbf{z}|\mathbf{x},\mathbf{y}) \parallel p(\mathbf{z})\big) - D_{KL}\big(q_{\phi_x}(\mathbf{z}|\mathbf{x}) \parallel p(\mathbf{z})\big) \\
& - D_{KL}\big(q_{\phi_y}(\mathbf{z}|\mathbf{y}) \parallel p(\mathbf{z})\big)
\end{aligned}
\tag{7}
$$

Where $\lambda_y^{yx}$ and $\lambda_y^y$ were hyperparameters. We set these to 10 and 100 respectively, following the reported best values from (Vedantam et al., 2017).

We trained the model using the frozen-likelihood trick shown to improve the model performance in Vedantam et al. (2017). The symbol decoder parameters $\theta_y$ were trained only using the $\mathbb{E}_{q_\phi(\mathbf{z}|\mathbf{x},\mathbf{y})} \left[\log p_{\theta_y}(\mathbf{y}|\mathbf{z})\right]$ term and not the $\mathbb{E}_{q_{\phi_y}(\mathbf{z}|\mathbf{y})} \left[\log p_{\theta_y}(\mathbf{y}|\mathbf{z})\right]$ term. For fair comparison to the other models, we did not utilise a product of experts for the inference network $q_{\phi_y}(\mathbf{z}|\mathbf{y})$.

In all architectural respects, the networks used were identical to those reported above for the JMVAE. The same training procedures were followed.

**Accuracy and diversity evaluation**   The classifier used to evaluate the samples generated by each model was trained to discriminate the four room configuration factors in the DeepMind Lab dataset: wall colour, floor colour, object colour and object identity. We used a network of four 2-strided deconvolutional layers (with filters in each successive layer of $\{32, 64, 128, 256\}$, and kernels sized 3x3), followed by a fully connected layer with 256 neurons, with ReLU activations used throughout. The output layer consisted of four fully connected softmax heads, one for each predicted factor (with dimensionality 16 for each of the colour factors, 3 for object identity). The classifier was trained until convergence using the Adam optimiser, with a learning rate of $1e-4$ and batch size of 100 (reaching an overall accuracy of 0.992).

The accuracy metric for the sym2img samples was computed as the average top-k accuracy across the factors (with $k = 3$ for the colour factors, and $k = 1$ for the object identity factor), against the ground-truth factors specified by the concept used to generate each sym2img sample. The top-k of the factors in each image sample was calculated using the top-k softmax outputs of the classifier.

Sample diversity of the sym2img data was characterised by estimating the KL divergence of the irrelevant factor distribution inferred for each concept with a flat distribution, $D_{KL}\big(u(\mathbf{y}_i) \parallel p(\mathbf{y}_i)\big)$. Here, $p(\mathbf{y}_i)$ is the joint distribution of the irrelevant factors in the sym2img set of images generated from the $i$th concept, which we estimated by averaging the classifier predictions across those images. $u(\mathbf{y}_i)$ is the desired (flat) joint distribution of the same factors (i.e., where each factor value has equal probability). We also computed the expected KL if $p(\mathbf{y}_i)$ were estimated using the samples drawn from the flat distribution $u(\mathbf{y}_i)$. We report the mean of this KL across all the k-grams. We used 64 sym2img samples per concept.

## A.2   DEEPMIND LAB DATASET DETAILS

**RGB to HSV conversion**   The majority of the data generative factors to be learnt in the DeepMind Lab dataset correspond to colours (floor, wall and object). We found that it was hard to learn disentangled representations of these data generative factors with $\beta$-VAE. We believe this is because $\beta$-VAE requires a degree of smoothness in pixel space when traversing a manifold for a particular data generative factor in order to correctly learn this factor (Higgins et al., 2017a). The intuitively smooth notion of colour, however, is disrupted in RGB space (see Fig. 8). Instead, the intuitive human notion of colour is more closely aligned with hue in HSV space. Hence we added a pre-processing step that converted the DeepMind Lab frames from the RGB to HSV space before training $\beta$-VAE. This conversion preserved the dimensionality of the frames, since both RGB and HSV require three channels. We found that this conversion enabled $\beta$-VAE to achieve good disentangling results.

**k-hot experiments**   Our DeepMind Lab (Beattie et al., 2016) dataset contained 73 frames per room, where the configuration of each room was randomly sampled from the outer product of the four data generative factors: object identity and colour, wall and floor colour (18,883 unique factor combinations). All models were trained using a randomly sampled subset of 133 concepts (with 10 example images per concept), 30 extra concepts were used for training the recombination operators (20 example images per concept) and a further set of 50 concepts

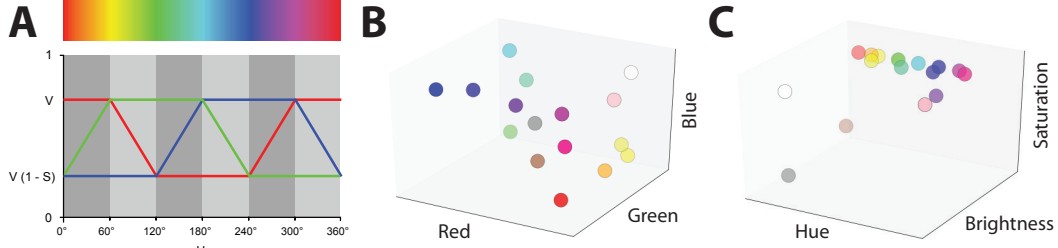

Figure 8: **A**: Comparison of hue traversal in HSV space, which closely aligns with the intuitive human understanding of colour, and the equivalent highly non-monotonic changes in RGB space. *H* stands for hue, *S* stands for saturation and *V* stands for value/brightness. Adapted from Wikipedia (2017). **B**: Visualisation of colours used in DeepMind Lab in RGB. **C**: Visualisation of colours used in DeepMind Lab in HSV. It can be seen that the HSV projection of the DeepMind Lab colours appears significantly more structured than the equivalent RGB projection.

were used to evaluate the models' ability to break away from their training distribution using recombination operators.

**Training the recombination operator**    The recombination operator was trained by sampling two concepts, $\mathbf{y}_1$ and $\mathbf{y}_2$, and an operator $\mathbf{r}$ as input. The training objective was to ground $\mathbf{z}_r$ in the ground truth latent space $\mathbf{z}_x$ inferred from an image $\mathbf{x}$. The ground truth image $\mathbf{x}$ was obtained by applying binary logical operation corresponding to $\mathbf{r}$ to binary symbols $\mathbf{y}_1$ and $\mathbf{y}_2$. This produces the ground truth recombined symbol $\mathbf{y}_r$, which can then be used to fetch a corresponding ground truth image $\mathbf{x}_r$ from the dataset.

To make sure that the logical operators were not presented with nonsensical instructions, we followed the following logic for sampling minibatches of $\mathbf{y}_1$ and $\mathbf{y}_2$ during training. The IN COMMON and AND operators were trained by sampling two k-grams $\mathbf{y}_1$ and $\mathbf{y}_2$ with $k \in \{1, 2, 3\}$. The IN COMMON operator had an additional restriction that the intersection cannot be and empty set. The IGNORE operator was trained by sampling a k-gram with $k \in \{1, 2, 3\}$ and a unigram selected from one of the factors specified by the k-gram.

### A.3    DEEPMIND LAB EXPERIMENTS

**Unsupervised visual representation learning**    SCAN relies on the presence of structured visual primitives. Hence, we first investigate whether $\beta$-VAE trained in an unsupervised manner on the visually complex DeepMind Lab dataset has discovered a disentangled representation of all its data generative factors. As can be seen in Fig. 9 (left panel), SCAN has learnt to represent each of the object-, wall-, and floor-colours, using two latents – one for hue and one for brightness. Learning a disentangled representation of colour is challenging, but we were able to achieve it by projecting the input images from RGB to HSV space, which is better aligned with human intuitions of colour (see Sec. A.2). We noticed that $\beta$-VAE confused certain colours (e.g. red floors are reconstructed as magenta, see the top right image in the Reconstructions pane of Fig. 9). We speculate that this is caused by trying to approximate the circular hue space using a linear latent. Red and magenta end up on the opposite ends of the linear latent while being neighbours in the circular space.

Compare the disentangled representations of Fig. 9 to the entangled equivalents in Fig. 10. Fig. 10 shows that an entangled $\beta$-VAE was able to reconstruct the data well, however due to the entangled nature of its learnt representations, latent traversal plots and samples are not as good as those of a disentangled $\beta$-VAE (Fig. 9).

**SCAN$_\text{U}$ analysis**    As shown in Fig. 10 SCAN with unstructured vision is based on a $\beta$-VAE that learnt a good (yet entangled) representation of the DeepMind Lab dataset. Due to the unstructured entangled nature of the visual latent space $\mathbf{z}_x$, the additional forward KL term of the SCAN loss function (Eq. 4) is not able to pick out the relevant visual primitives for each training concept. Instead, all latents end up in the irrelevant set, since the relevant and irrelevant ground truth factors end up being entangled in the latent space $\mathbf{z}_x$. This disrupts the ability of SCAN with entangled vision to learn useful concepts, as demonstrated in Fig. 11.

**JMVAE analysis**    In this section we provide some insights into the nature of representations learnt by JMVAE (Suzuki et al., 2017). Fig. 12 demonstrates that after training JMVAE is capable of reconstructing the data and drawing reasonable visual samples. Furthermore, the latent traversal plots indicate that the model learnt a reasonably disentangled representation of the data generative factors. Apart from failing to learn a latent to represent the spawn animation and a latent to represent all object identities (while the hat and the ice lolly are represented, the suitcase is missing), the representations learnt by JMVAE match those learnt by $\beta$-VAE (compare Figs. 9 and 12). Note, however, that unlike $\beta$-VAE that managed to discover and learn a disentangled

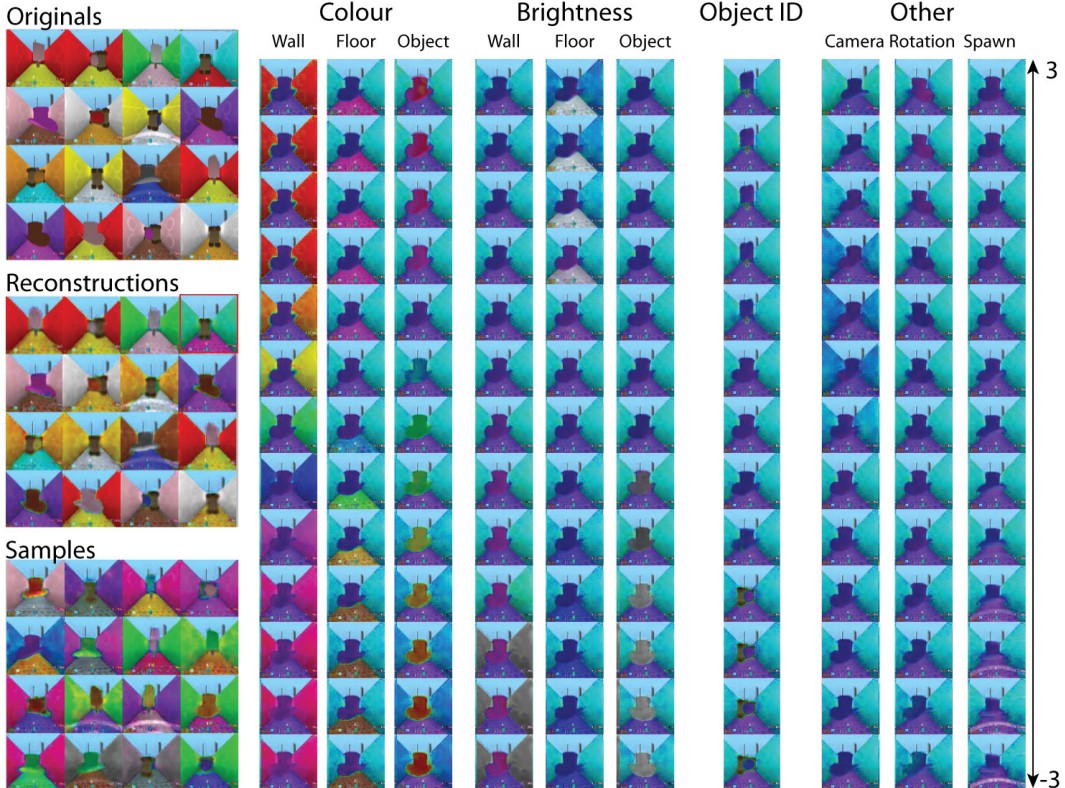

Figure 9: Reconstructions, samples and latent traversals of $\beta$-VAE ($\beta = 53$) trained to disentangle the data generative factors of variation within the DeepMind Lab dataset. For the latent traversal plots we sampled the posterior, then visualised $\beta$-VAE reconstructions while resampling each latent unit one at a time in the $[-3, 3]$ range while keeping all other latents fixed to their originally sampled values. This process helps visualise which data generative factor each latent unit has learnt to represent.

representation of the data generative factors in a completely unsupervised manner, JMVAE was able to achieve its disentangling performance by exploiting the extra supervision signal coming from the symbolic inputs.

JMVAE is unable to learn a hierarchical compositional latent representation of concepts like SCAN does. Instead, it learns a flat representation of visual primitives like the representation learnt by $\beta$-VAE. Such a flat representation is problematic, as evidenced by the accuracy/diversity metrics shown in Tbl. 1. Further evidence comes from examining the sym2img samples produced by JMVAE (see Fig. 13). It can be seen that JMVAE fails to learn the abstract concepts as defined in Sec. 3. While the samples in Fig. 13 mostly include correct wall colours that match their respective input symbols, the samples have limited diversity. Many samples are exact copies of each other – a sign of mode collapse.

**TrELBO analysis** This section examines the nature of representations learnt by TrELBO (Vedantam et al., 2017). Fig. 14 demonstrates that after training TrELBO is capable of reconstructing the data, however it produces poor samples. This is due to the highly entangled nature of its learnt representation, as also evidenced by the traversal plots. Since TrELBO is not able to learn a compositional latent representation of concept like that acquired by SCAN, it also struggles to produce diverse sym2img samples when instructed with symbols from the training set (see Fig. 15). Furthermore, this lack of structure in the learnt concept representations precludes successful recombination operator training. Hence, sym2img samples of test symbols instructed through recombination operators lack accuracy (Fig. 16).

**Data efficiency analysis** We evaluate the effect of the training set size on the performance of SCAN, JMVAE and TrELBO by comparing their accuracy and diversity scores after training on {5, 10, 15, 20, 25, 50, 75} concepts. Fig. 17 shows that SCAN consistently outperforms its baselines in terms of the absolute scores, while also displaying less variance when trained on datasets of various sizes. For this set of experiments we also halved the number of training iterations for all models, which affected the baselines but not SCAN. The diversity of JMVAE and TrELBO is better in this plot compared to the results reported in Tbl. 1 because sym2img samples used for this plot were blurrier than those describe in the main text.

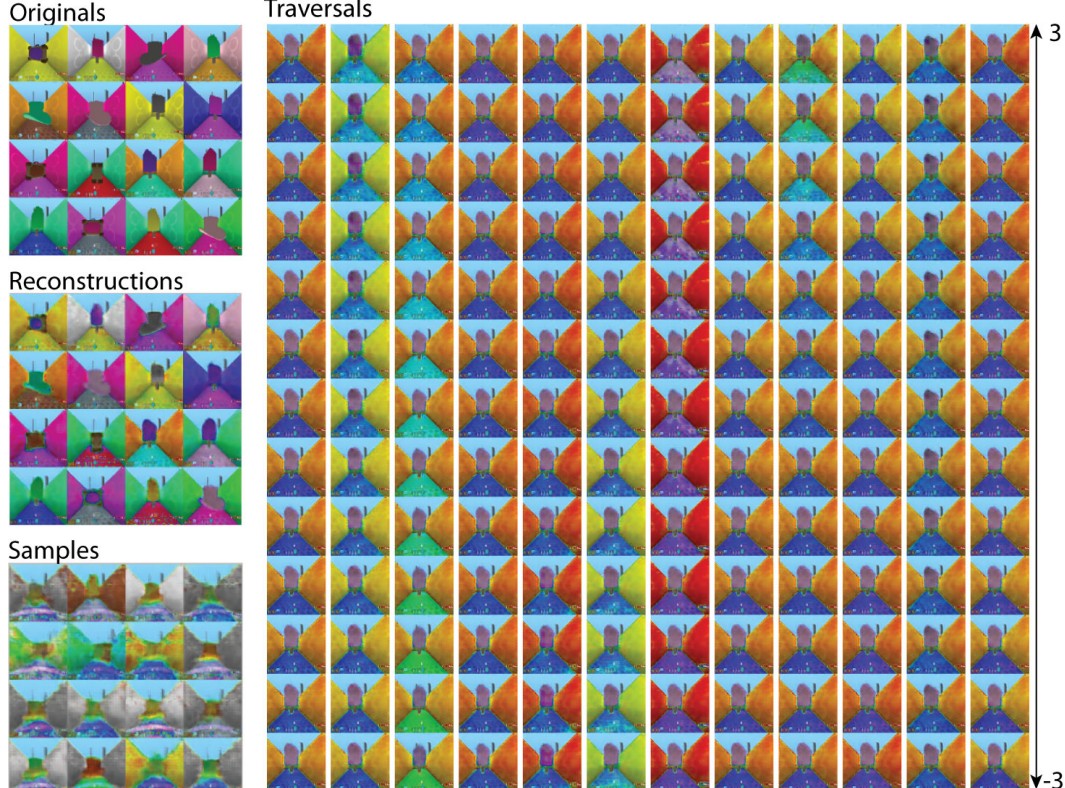

Figure 10: Samples, reconstructions and latent traversals of $\beta$-VAE that did not learn a structured disentangled representation ($\beta = 0.1$). It is evident that the model learnt to reconstruct the data despite learning an entangled latent space.

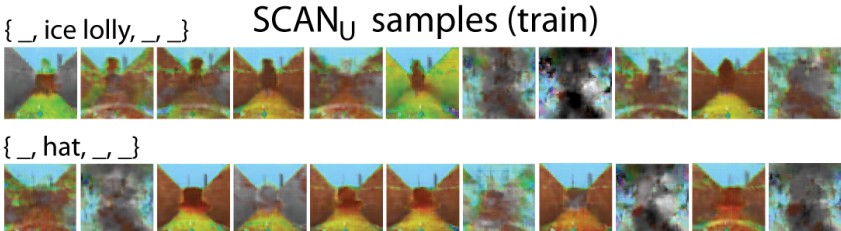

Figure 11: Visual samples (sym2img) of SCAN grounded in unstructured vision when presented with symbols "hat" and "ice lolly". It is evident that the model struggled to learn a good understanding of the meaning of these concepts.

## A.4 DSPRITES EXPERIMENTS

In this section we describe additional experiments testing SCAN on the dSprites (Matthey et al., 2017) dataset. The dataset consists of binary sprites fully specified by five ground truth factors: position x (32 values), position y (32 values), scale (6 values), rotation (40 values) and sprite identity (3 values). For our experiments we defined a conceptual space spanned by three of the data generative factors - horizontal and vertical positions, and scale. We quantised the values of each chosen factor into halves (top/bottom, left/right, big/small) and assigned one-hot encoded symbols to each of the $\sum_{k=1}^{K} \binom{K}{k} N^k = 26$ possible concepts to be learnt (since $K = 3$ is the number of factors to be learnt and $N = 2$ is the number of values each factor can take). We compared the performance of SCAN grounded in disentangled visual representations ($\beta$-VAE with $\beta = 12$) to that of grounded in entangled visual representations ($\beta$-VAE with $\beta = 0$). We trained both models on a random subset of image-symbol pairs $(x_i, y_i)$ making up $< 0.01\%$ of the full dataset.

We quantified how well the models understood the meaning of the positional and scale concepts after training by running sym2img inference and counting the number of white pixels within each of the four quadrants of the canvas (for position) or in total in the whole image (for scale). This can be compared to similar values calculated

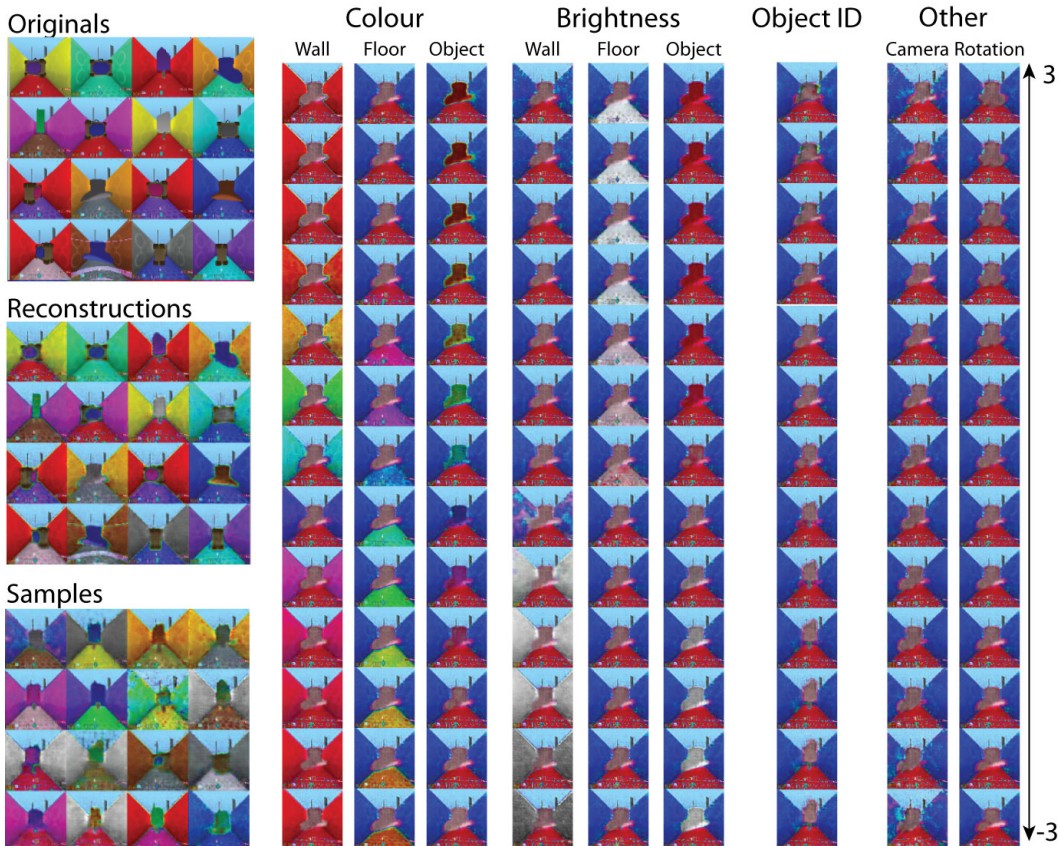

Figure 12: Samples, reconstructions and latent traversals of JMVAE. The model learns good disentangled latents, making use of the supervised symbolic information available.

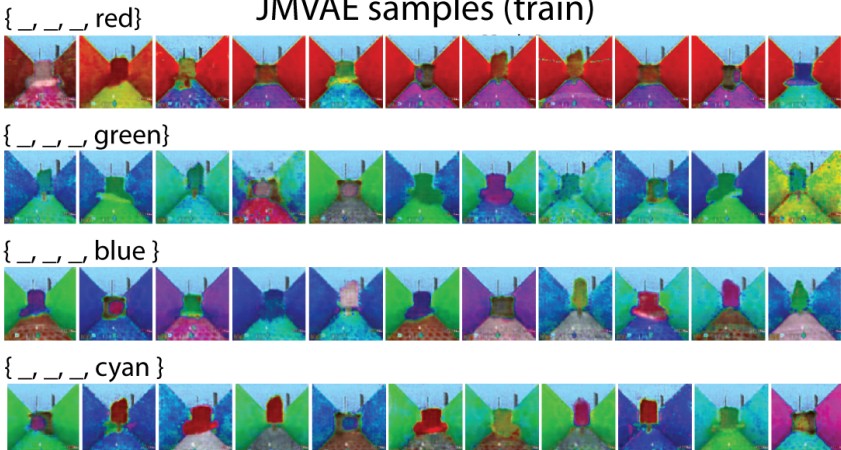

Figure 13: Visualisation of sym2img visual samples produced JMVAE in response to symbols specifying wall colour names. It is evident that the model suffers from mode collapse, since a significant number of samples are copies of each other.

over a batch of ground truth images that match the same input symbols. Samples from SCAN matched closely the statistics of the ground truth samples (see Fig. 18). SCAN$_U$, however, failed to produce meaningful samples despite being able to reconstruct the dataset almost perfectly.

## A.5 CELEBA EXPERIMENTS

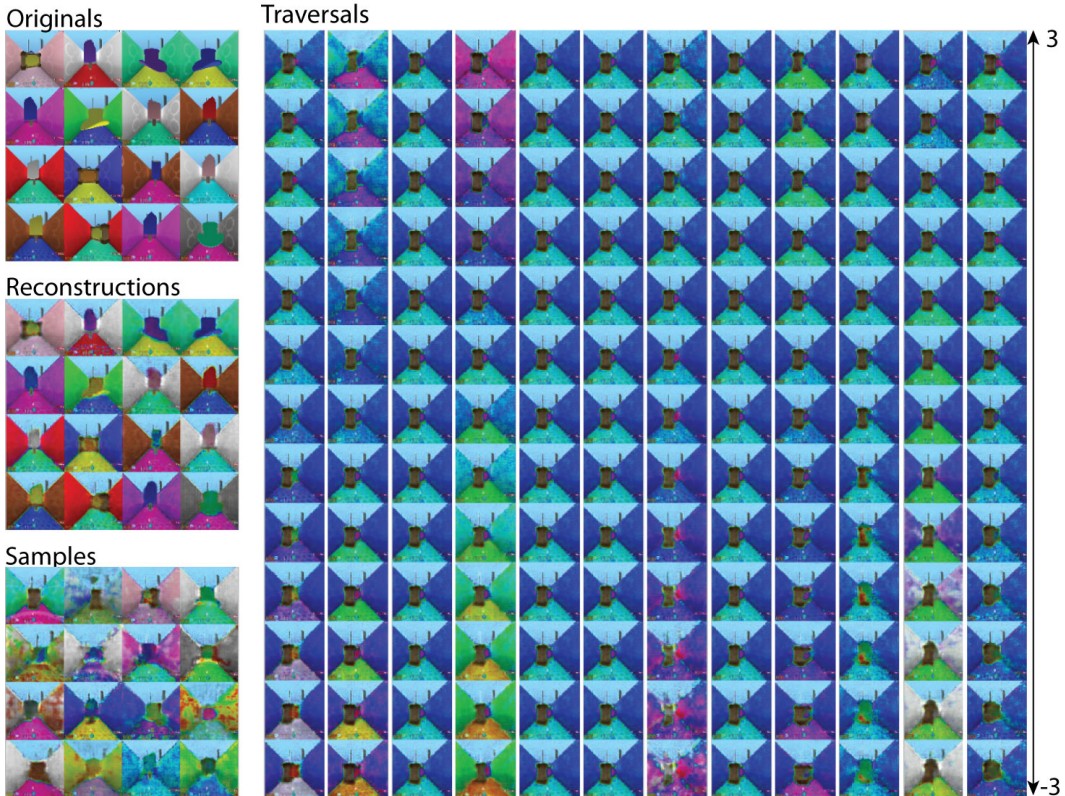

Figure 14: Samples, reconstructions and latent traversals of TrELBO. The model learns a very entangled representation.

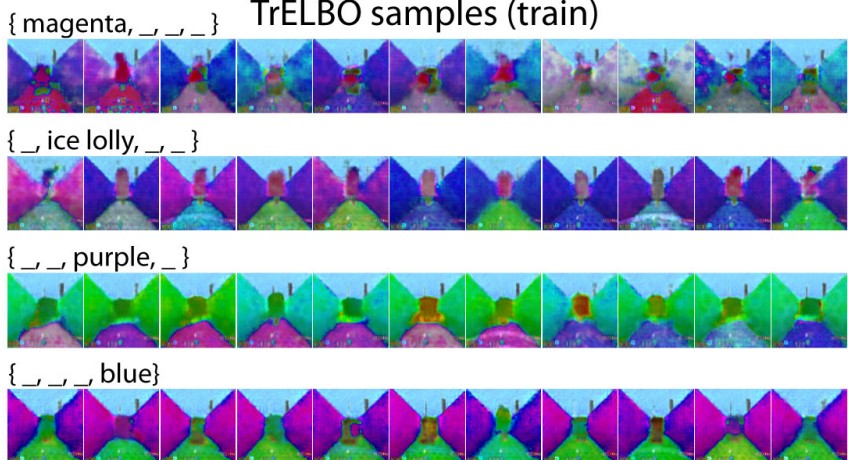

Figure 15: Visualisation of sym2img visual samples produced TrELBO in response to train symbols: "magenta object", "ice lolly", "purple floor" and "blue wall". It is evident that the model has good accuracy but very low diversity.

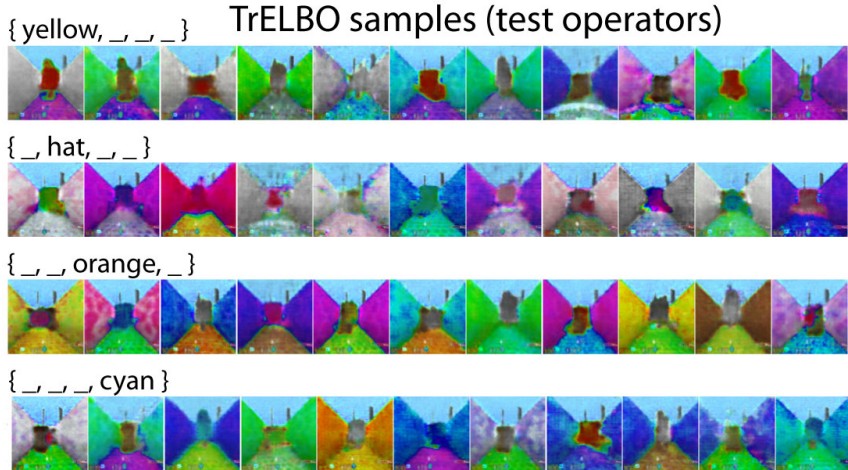

Figure 16: Visualisation of sym2img visual samples produced TrELBO in response to test symbols instructed using recombination operators: "yellow object", "hat", "orange floor" and "cyan wall". It is evident that the model has very low accuracy but decent diversity.

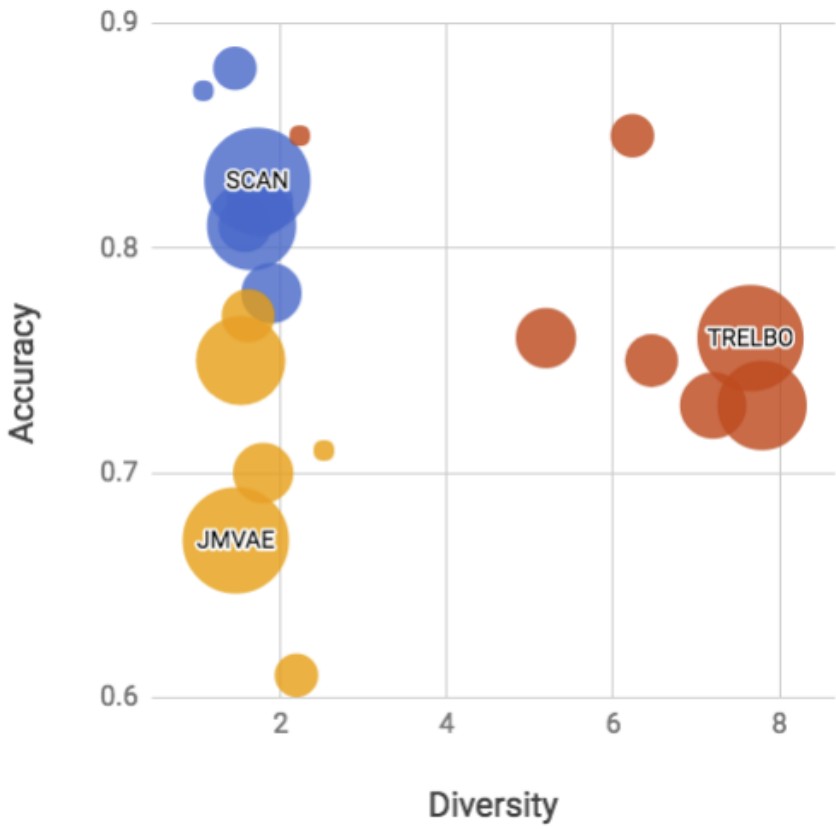

Figure 17: Accuracy and diversity scores of SCAN, JMVAE and TrELBO after being trained on {5, 10, 15, 20, 25, 50, 75} concepts with 10 visual examples each. The size of the circle corresponds to the training set size. We used symbols from the train set to generate sym2img samples used to calculate the scores. SCAN outperforms both baselines and shows less susceptibility to the training set size.

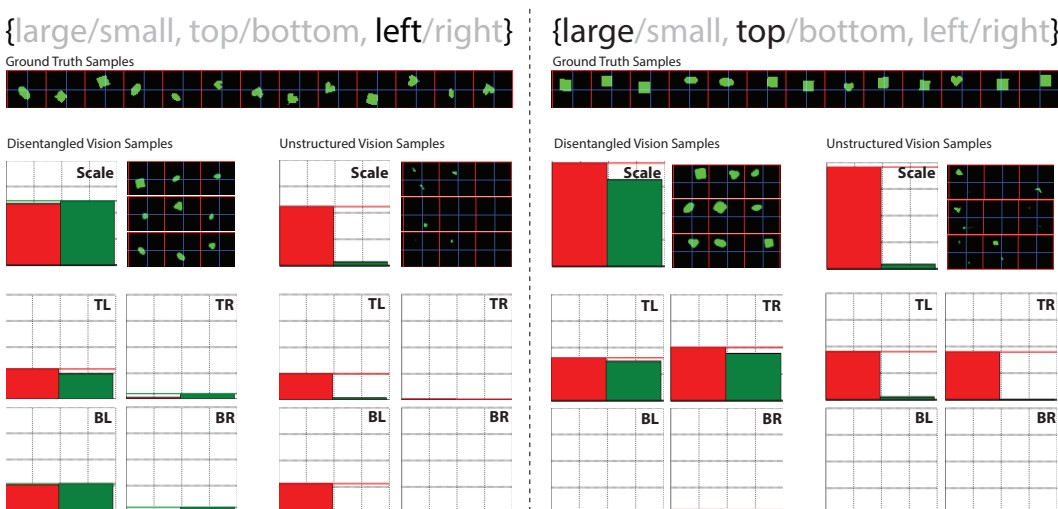

Figure 18: sym2img inference performance of SCAN and SCAN$_U$ for symbols - "left" and "large top". First line in each subplot demonstrates ground truth samples from dSprites dataset that correspond to the respective symbol. Next three lines illustrate the comparative performance of SCAN (left) vs SCAN$_U$ (right), including their respective sym2img samples, as well as the quantitative comparison of each model (green) to the ground truth (red) in terms of scale understanding (each bar corresponds to the average number of pixels per sample image) and positional understanding (each bar corresponds to the average number of pixels in one of the four quadrants of the samples: T - top, B - bottom, R - right, L - left). The closer the green bars are to the red bars, the better the model's understanding of the learnt concepts.

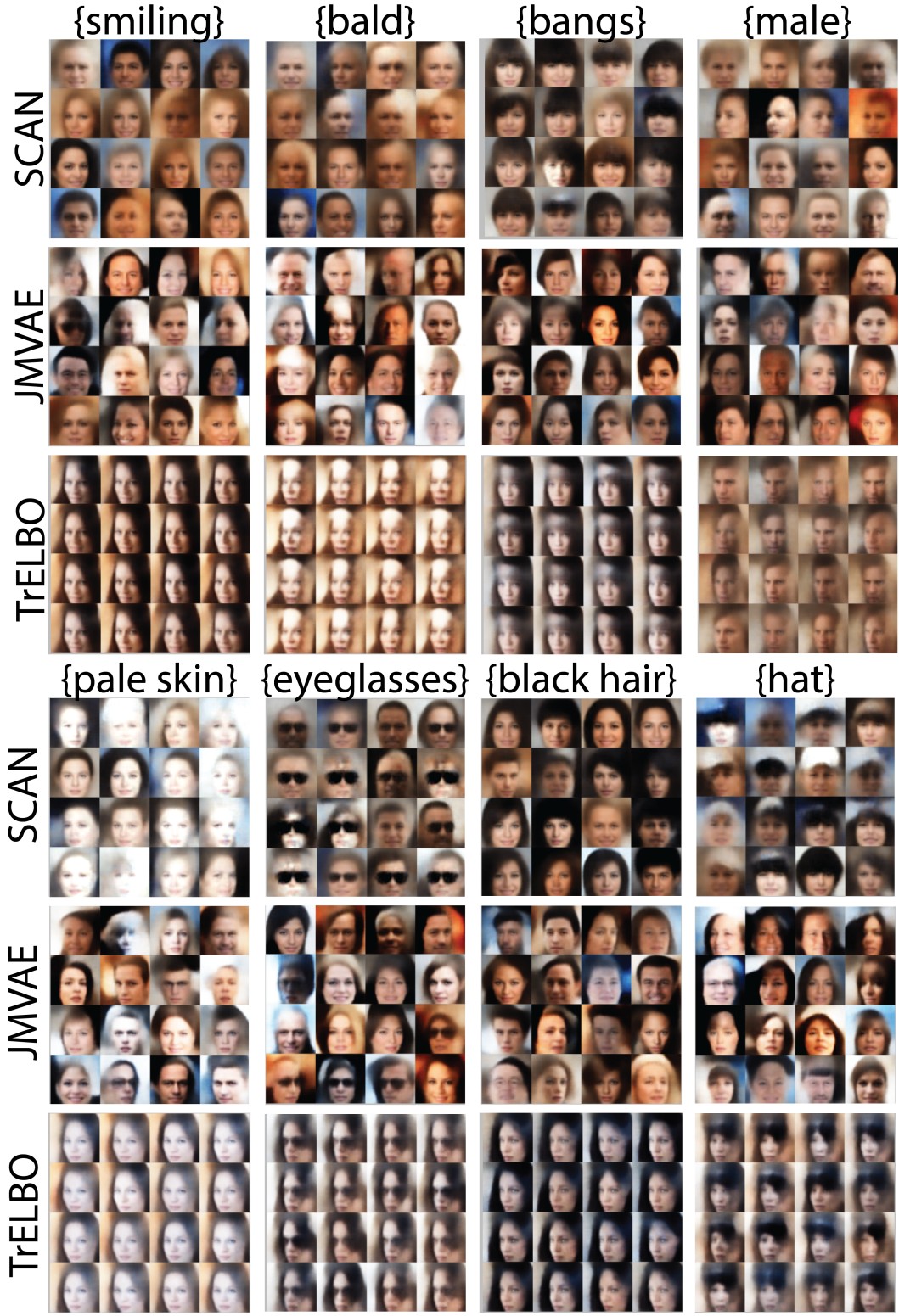

Figure 19: Large version of Fig. 6

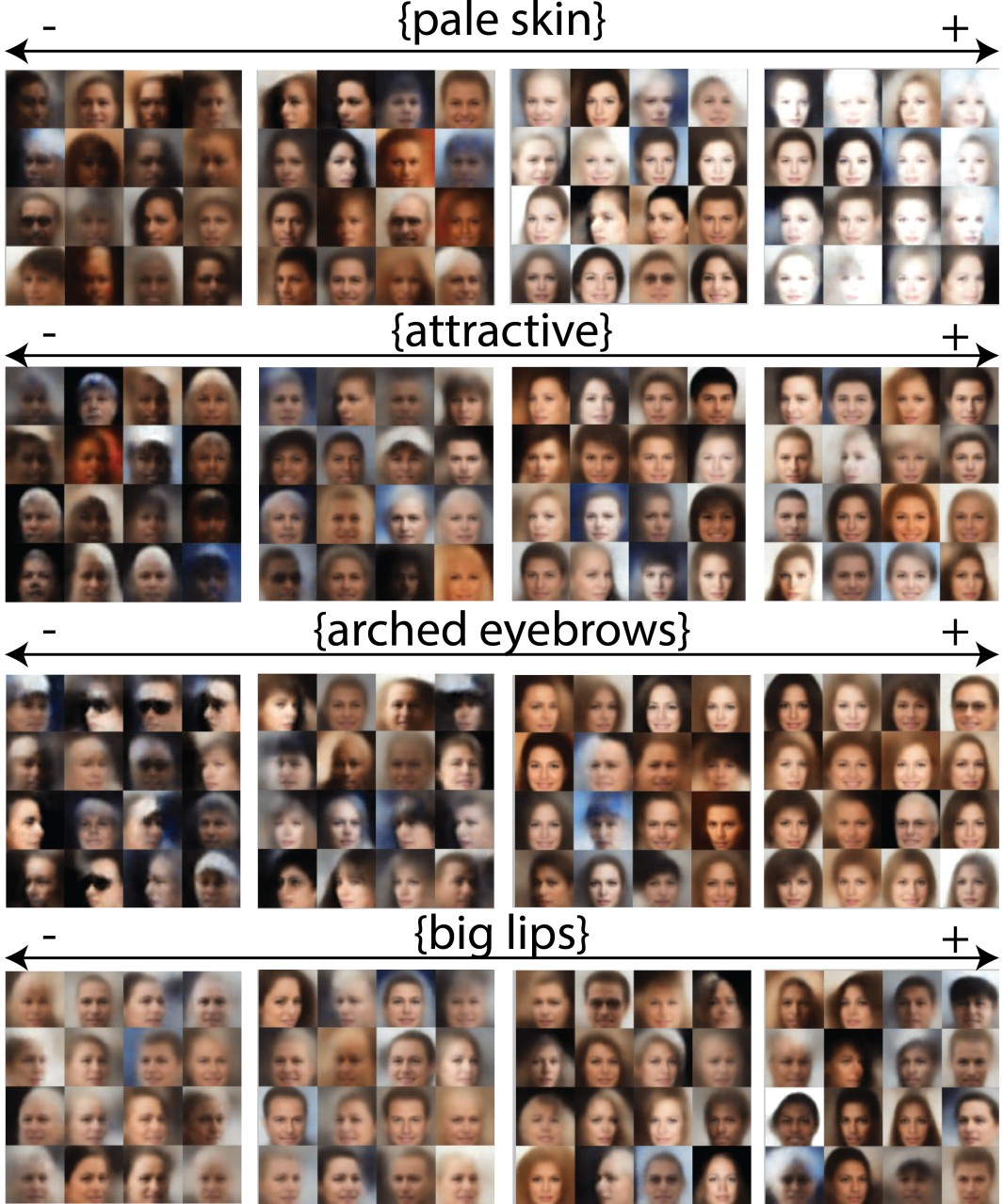

Figure 20: Large version of Fig. 7

