# OpenReview forum: "SCAN: Learning Hierarchical Compositional Visual Concepts"
_ICLR.cc/2018/Conference — Accept (Poster)_

### Official Review · AnonReviewer3 · 2017-11-29
**A neural network that learns visual concepts and basic operators over them.**

**Rating:** 5
**Confidence:** 4

**Review:**

This paper proposed a novel neural net architecture that learns object concepts by combining a beta-VAE and SCAN. The SCAN is actually another beta-VAE with an additional term that minimizes the KL between the distribution of its latent representation and the first beta-VAE’s latent distribution. The authors also explored how this structure could be further expanded to incorporate another neural net that learns operators (and, in common, ignore), and demonstrated that the proposed system is able to generate accurate and diverse scenes given the visual descriptions.

In general, I think this paper is interesting. It’s studying an important problem with a newly proposed neural net structure. The experimental results are good and the model is compared with very recent baselines.

I am, however, still lukewarm on this submission for its limited technical innovation and over-simplified experimental setup.

This paper does have technical innovations: the SCAN architecture and the way they learn “recombination operators” are newly proposed. However, there are in essence very straightforward extensions of VAE and beta-VAE (this is based on the fact that beta-VAE itself is a simple modification of VAE and the effect was discussed in a number of concurrent papers).

This would still be fine, as many small modifications of neural net architecture turn out to reveal fundamental insights that push the field forward. This is, however, not the case in this paper (at least not in the current manuscript) due to its over-simplified experiments. The authors are using images as input, but the images are all synthetic, and further, they are all synthesized to have highly regular structure. This suggests the network is likely to overfit the data and learn a straightforward mapping from input to the code. It’s unclear how well the system is able to generalize to real-world scenarios. Note that even datasets like MNIST has much higher complexity than the dataset used in this paper (though the dataset in this paper is more colorful).

I agree that the proposed method performs better that its recent competitors. However, many of those methods like TripleELBO are not explicitly designed for these ‘recombination operators’. In contrast, they seem to perform well on real datasets. I would strongly suggest the authors perform additional experiments on standard benchmarks for a fair comparison.

---

> ### Author Response · Authors · 2017-12-22
> **Extra experiments with CelebA demonstrate that SCAN significantly outperforms JMVAE and TrELBO**
>
> Dear Reviewer,
>
> Thank you for your feedback. We have added an additional section describing the comparison of our approach to JMVAE and TrELBO on CelebA. Unlike the similar TrELBO experiments, we did minimal pre-processing of the dataset (only cropping to 64x64) and trained the models on the noisy attribute labels out of the box. As you may be aware, CelebA attributes are notoriously unreliable - many are subjective, refer to aspects of the images that get cropped away or are plain wrong. Our experiments demonstrate that SCAN significantly outperforms both baselines (but TrELBO in particular) and discovers a subset of attributes that refer to something meaningful based on the visual examples present in the dataset, while ignoring the uninformative attributes. SCAN is then able to traverse the individual directions of variation it has discovered and imagine both positive and negative examples of the attribute. This is unlike the baselines, which can only imagine positive examples after being trained on positive examples.
>
> We hope that our experiments address your  concerns about the technical innovation of our approach, since we demonstrate that currently SCAN is the only model that is able to learn compositional hierarchical visual concepts on real visual datasets.
>
> Happy holidays!

---

> > ### Comment · AnonReviewer3 · 2018-01-24
> > **post-rebuttal comments**
> >
> > I appreciate the authors' effort along the direction. The additional experiments strengthened the paper, but I feel it still needs more work.
> >
> > The technical innovation of the paper is to learn 'recombination operators'. As I said in the original review, methodologically the innovation is quite straightforward, but it can make a good paper if well evaluated. The additional experiments on celebA, however, are not evaluating the 'recombination operators'. It is basically suggesting beta-VAE can learn smooth interpolations (or extrapolations) given a certain attribute. This is nice, but connects better to the original beta-VAE paper than to this paper.
> >
> > In general, this paper has great potentials but will benefit from another cycle. Would that be possible to really learn recombination operators on real images? If SCAN can learn concepts like pale-skin (and) big lips, or attractive (ignore) arched eyebrows, the paper will be much stronger.

---

> > > ### Author Response · Authors · 2018-01-25
> > > **Response to post-rebuttal comments**
> > >
> > > Dear Reviewer,
> > >
> > > Thank you for taking the time to comment on the updated version of our paper. You suggest that you do not find our additional experiments convincing enough because we do not train recombination operators on the celebA dataset. However, in our understanding your original review did not ask for these experiments. It suggested that we do a fair comparison with the JMVAE and TrELBO baselines on a real dataset, followed by a remark that the baselines were not explicitly designed for recombination operators. In our understanding it implied that the only fair comparison was to compare the abstract concept learning step across the original models. Furthermore, it is unfortunate that your request for the additional experiments with the recombination operators has arrived at this stage. While we cannot update our manuscript before the decision deadline, we would be happy to run the additional experiments for the camera ready version of the paper.
> > >
> > > Your original review had reservations about the technical novelty of our approach, which you stated in itself was not a problem as long as we could demonstrate that our approach outperforms the current state of the art methods on realistic datasets. We believe that our new experiments on CelebA demonstrate exactly that.
> > >
> > > In your current comment you suggest that our additional CelebA experiments only demonstrate that beta-VAE can learn smooth interpolations and extrapolations of certain attributes. However, we believe that our additional experiments demonstrate that SCAN can learn new meaningful abstractions that are grounded in the basic visual factors discovered by beta-VAE, but which beta-VAE alone could not have, and in fact did not discover.
> > >
> > > In addition, please note that unlike the CelebA experiments in the TrELBO paper, we did not remove mislabeled attributes from the training set, which consequently made the training task significantly harder for all models. The fact that SCAN was able to work well in such a setting is a further demonstration of the robustness of our approach.
> > >
> > > In summary, we believe that we have demonstrated the usefulness and the power of our approach over the recent state of the art baseline methods on an important problem of learning hierarchical compositional visual concepts. Our approach may seem -- at first glance -- like a “straightforward” modification to existing VAE variants, but it is the only one that is currently able to discover meaningful compositional visual abstractions on realistic datasets.

---

### Official Review · AnonReviewer1 · 2017-12-03
**interesting idea, but limited experimental evaluation**

**Rating:** 6
**Confidence:** 4

**Review:**

This paper introduces a VAE-based model for translating between images and text. The main way that their model differs from other multimodal methods is that their latent representation is well-suited to applying symbolic operations, such as AND and IGNORE, to the text. This gives them a more expressive language for sampling images from text.

Pros:
- The paper is well written, and it provides useful visualizations and implementation details in the appendix.

- The idea of learning compositional representations inside of a VAE framework is very appealing.

- They provide a modular way of learning recombination operations.

Cons:
- The experimental evaluation is limited. They test their model only on a simple, artificial dataset. It would also be helpful to see a more extensive evaluation of the model's ability to learn logical recombination operators, since this is their main contribution.

- The approach relies on first learning a pretrained visual VAE model, but it is unclear how robust this is. Should we expect visual VAEs to learn features that map closely to the visual concepts that appear in the text? What happens if the visual model doesn't learn such a representation? This again could be addressed with experiments on more challenging datasets.

- The paper should explain the differences and trade offs between other multimodal VAE models (such as their baselines, JMVAE and TrELBO) more clearly. It should also clarify differences between the SCAN_U baseline and SCAN in the main text.

- The paper suggests that using the forward KL-divergence is important, but this does not seem to be tested with experiments.

- The three operators (AND, IN COMMON, and IGNORE) can easily be implemented as simple transformations of a (binary) bag-of-words representation. What about more complex operations, such as OR, which seemingly cannot be encoded this way?

Overall, I am borderline on this paper, due to the limited experimental evaluation, but lean slightly towards acceptance.

---

> ### Author Response · Authors · 2017-12-22
> **We have added extra experiments with CelebA, SCAN_U and SCAN with reverse KL and show that SCAN still significantly outperforms all baselines**
>
> Dear Reviewer,
>
> Thank you for your feedback. Please find the responses to your points below:
>
>
> - The experimental evaluation is limited. They test their model only on a simple, artificial dataset. It would also be helpful to see a more extensive evaluation of the model's ability to learn logical recombination operators, since this is their main contribution.
>
> We have now added an additional section demonstrating that SCAN significantly outperforms both JMVAE and TrELBO on CelebA - a significantly more challenging and realistic dataset.
>
>
>
>
> - The approach relies on first learning a pretrained visual VAE model, but it is unclear how robust this is. Should we expect visual VAEs to learn features that map closely to the visual concepts that appear in the text? What happens if the visual model doesn't learn such a representation? This again could be addressed with experiments on more challenging datasets.
>
> SCAN does indeed rely on learning disentangled visual representations as defined in Bengio (2013) and Higgins et al (2017). The performance of SCAN drops as the quality of disentanglement drops, as demonstrated by the additional SCAN_U baselines we have added to Table 1. It has, however, been shown that beta-VAE is able to learn disentangled representation on more challenging datasets (Higgins et al, 2017a, b), and we have shown that SCAN can significantly outperform both JMVAE and TrELBO on CelebA in the additional section we have added at the end of the paper. When training SCAN on CelebA, we show that SCAN is able to ignore symbolic (text) attributes that do not refer to anything meaningful in the image space, and ground the remaining attributes in whatever dictionary of visual primitives it has access to (not all of which map directly to the symbolic attributes). For example, the “attractiveness” attribute is subjective and has no direct mapping to a particular visual primitive, yet SCAN learns that in the CelebA dataset it tends to refer to young females.
>
>
>
>
>
> - The paper should explain the differences and trade offs between other multimodal VAE models (such as their baselines, JMVAE and TrELBO) more clearly. It should also clarify differences between the SCAN_U baseline and SCAN in the main text.
>
> We have added the explanations in text. In summary, TrELBO tends to learn a flat and unstructured conceptual latent space, that results in very poor diversity of their samples. JMVAE, on the other hand, comes close to our approach in the limit where the text labels provide enough supervision to help disentangle the joint latent space q(z|x,y). In that case, the joint posterior q(z|x,y) and the symbolic posterior q(z|y) of JMVAE become equivalent to the visual posterior q(z|x) and symbolic posterior q(z|y) of SCAN, since both use forward KL to ground q(z|y). Hence, the biggest differences between our approach and JMVAE are: 1) we are able to learn disentangled visual primitives in an unsupervised manner while JMVAE relies on good structured labels to supervise this process; 2) we use a staged optimisation process, where we first learn vision, then concepts, while JMVAE performs joint optimisation. In practice we find that JMVAE training is more sensitive to architectural and hyperparameter choices and hence most of the time performs worse than SCAN.
>
> SCAN_U is a version of SCAN that grounds concepts in an unstructured visual latent space. We have now added extra experiments to show how the performance of SCAN drops as the level of visual disentanglement in SCAN_U is decreased.
>
>
>
>
> - The paper suggests that using the forward KL-divergence is important, but this does not seem to be tested with experiments.
>
> We have added the additional baseline with reverse KL (SCAN_R) to Table 1 and showed that it has really bad diversity as predicted by our reasoning.
>
>
>
>
> - The three operators (AND, IN COMMON, and IGNORE) can easily be implemented as simple transformations of a (binary) bag-of-words representation. What about more complex operations, such as OR, which seemingly cannot be encoded this way?
>
> In this work, we focus on operators that can be used to traverse the implicit hierarchy of concepts, and since OR is not one of such operators, it is outside the scope of the current paper. We agree that it is interesting to implement and study additional, more complex operations, which we leave for future work.
>
> Happy holidays!

---

### Official Review · AnonReviewer2 · 2017-12-04
**Good paper, but some pieces are missing**

**Rating:** 7
**Confidence:** 4

**Review:**

Summary
---
This paper proposes a new model called SCAN (Symbol-Concept Association Network) for hierarchical concept learning. It trains one VAE on images then another one on symbols and aligns their latent spaces. This allows for symbol2image and image2symbol inference. But it also allows for generalization to new concepts composed from existing concepts using logical operators. Experiments show that SCAN generates images which correspond to provided concept labels and span the space of concepts which match these labels.

The model starts with a beta-VAE trained on images (x) from the relevant domain (in this case, simple scenes generated from DeepMind Lab which vary across a few known dimensions). This is complemented by the SCAN model, which is a beta-VAE trained to reconstruct symbols (y; k-hot encoded concepts like {red, suitcase}) with a slightly modified objective. SCAN optimizes the ELBO plus a KL term which pushes the latent distribution of the y VAE toward the latent distribution of the x (image) VAE. This aligns the latent representations so now a symbol can be encoded into a latent distribution z and decoded as an image.

One nice property of the learned latent representation is that more specific concepts have more specific latent representations. Consider latent distributions z1 and z2 for a more general symbol {red} and a more specific symbol {red, suitcase}. Fewer dimensions of z2 have high variance than dimensions of z1. For example, the latent space could encode red and suitcase in two dimensions (as binary attributes). z1 would have high variance on all dimensions but the one which encodes red and z2 would have high variance on all dimensions but red and suitcase. In the reported experiments some of the dimensions do seem to be interpretable attributes (figure 5 right).

SCAN also pays particular attention to hierarchical concepts. Another very simple model (1d convolution layer) is learned to mimic logical operators. Normally a SCAN encoder takes {red} as input and the decoder reconstructs {red}. Now another model is trained that takes "{red} AND {suitcase}" as input and reconstructs {red, suitcase}. The two input concepts {red} and {suitcase} are each encoded by a pre-trained SCAN encoder and then those two distributions are combined into one by a simple 1d convolution module trained to implement the AND operator (or IGNORE/IN COMMON). This allows images of concepts like {small, red, suitcase} to be generated even if small red suitcases are not in the training data.

Experiments provide some basic verification and analysis of the method:
1) Qualitatively, concept samples are correct and diverse, generating images with all configurations of attributes not specified by the input concept.
2) As SCAN sees more diverse examples of a concept (e.g. suitcases of all colors instead of just red ones) it starts to generate more diverse image samples of that concept.
3) SCAN samples/representations are more accurate (generate images of the right concept) and more diverse (far from a uniform prior in a KL sense) than JMVAE and TELBO baselines.
4) SCAN is also compared to SCAN_U, which uses an image beta-VAE that learned an entangled (Unstructured) representation. SCAN_U performed worse than SCAN
and baselines.
5) Concepts expressed as logical combinations of other concepts generalize well for both the SCAN representation and the baseline representations.


Strengths
---

The idea of concept learning considered here is novel and satisfying. It imposing logical, hierarchical structure on latent representations in a general way. This suggests opportunities for inserting prior information and adds interpretability to the latent space.


Weaknesses
---

I think this paper is missing some important evaluation.

Role/Nature of Disentangled Features not Clear (major):

* Disentangled features seem to be very important for SCAN to work well (SCAN vs SCAN_U). It seems that the only difference between the unstructured (entangled) and the structured (disentangled) visual VAE is the color space of the input (RGB vs HSV). If so, this should be stated more clearly in the main paper. What role did beta-VAE (tuning beta) as opposed to plain VAE play in learning disentangled features?

* What color space was used for the JMVAE and TELBO baselines? Training these with HSV seems especially important for establishing a good comparison, but it would be good to report results for HSV and RGB for all models.

* How specific is the HSV trick to this domain? Would it matter for natural images?

* How would a latent representation learned via supervision perform? (Maybe explicitly align dimensions of z to red/suitcase/small with supervision through some mechanism. c.f. "Discovering Hidden Factors of Variation in Deep Networks" by Cheung et al.)

Evaluation of sample complexity (major):

* One of the main benefits of SCAN is that it works with less training data. There should be a more systematic evaluation of this claim. In particular, I would like to see a Number of Examples vs Performance (Accuracy/Diversity) plot for both SCAN and the baselines.

Minor questions/comments/concerns:

* What do the logical operators learn that the hand-specified versions do not?

* Does training SCAN with the structure provided by the logical operators lead to improved performance?

* There seems to be a mistake in figure 5 unless I interpreted it incorrectly. The right side doesn't match the left side. During the middle stage of training object hues vary on the left, but floor color becomes less specific on the right. Shouldn't object color become less specific?


Prelimary Evaluation
---

This clear and well written paper describes an interesting and novel way of learning a model of hierarchical concepts. It's missing some evaluation that would help establish the sample complexity benefit more precisely (a claimed contribution) and add important details about unsupervised disentangled representations. I would happy to increase my rating if these are addressed.

---

> ### Author Response · Authors · 2017-12-22
> **We have added sample complexity evaluation that demonstrates that SCAN training is more stable than JMVAE and TrELBO, our experiments with CelebA in RGB space clarify the role of colour space in learning disentangled representations**
>
> Dear Reviewer,
>
> Thank you for your feedback. Please find the responses to your points below:
>
> Role/Nature of Disentangled Features not Clear (major):
>
> * Disentangled features seem to be very important for SCAN to work well (SCAN vs SCAN_U). It seems that the only difference between the unstructured (entangled) and the structured (disentangled) visual VAE is the color space of the input (RGB vs HSV). If so, this should be stated more clearly in the main paper. What role did beta-VAE (tuning beta) as opposed to plain VAE play in learning disentangled features?
>
> The statement about the “only difference”  is not quite right. While an HSV colour space helps beta-VAE disentangle the particular DeepMind Lab dataset we used, the conversion from RGB to HSV is not sufficient for disentangling. As shown in our additional SCAN_U experiments in Table 1, it is still important to use a carefully tuned beta-VAE rather than a plain VAE to get good enough disentanglement for SCAN to work. Furthermore, we have added additional experiments with CelebA where we learn disentangled visual representations with a beta-VAE in RGB space. A plain VAE is unable to learn such disentangled representations, as was shown in Higgins et al, 2017.
>
>
>
>
>
> * What color space was used for the JMVAE and TELBO baselines? Training these with HSV seems especially important for establishing a good comparison, but it would be good to report results for HSV and RGB for all models.
>
> All baselines are trained in HSV space when using the DeepMind Lab dataset in our paper. We have now added additional experiments on CelebA, where all models are now trained using the RGB colour space.
>
>
>
>
>
> * How specific is the HSV trick to this domain? Would it matter for natural images?
>
> The HSV trick was useful for the DeepMind Lab dataset, but it is not necessary for all datasets as demonstrated in the new CelebA experiments.
>
>
>
>
>
> * How would a latent representation learned via supervision perform? (Maybe explicitly align dimensions of z to red/suitcase/small with supervision through some mechanism. c.f. "Discovering Hidden Factors of Variation in Deep Networks" by Cheung et al.)
>
> A latent representation learnt via supervision would also work, as long as the latent distribution is from the location/scale distributional family. Hence, the work by Cheung et al or DC-IGN by Kulkarni et al would both be suitable for grounding SCAN. We concentrated on the unsupervised beta-VAE, since we wanted to minimise human intervention and bias.
>
>
>
>
>
> Evaluation of sample complexity (major):
>
> * One of the main benefits of SCAN is that it works with less training data. There should be a more systematic evaluation of this claim. In particular, I would like to see a Number of Examples vs Performance (Accuracy/Diversity) plot for both SCAN and the baselines.
>
> We have added a plot with this information in the supplementary materials.
>
>
>
>
>
> Minor questions/comments/concerns:
>
> * What do the logical operators learn that the hand-specified versions do not?
>
> In general we find that the learnt operators have better accuracy and diversity, achieving 0.79 (learnt) vs 0.54 (hand crafted) accuracy (higher is better) and 1.05 (learnt) vs 2.03 (hand crafted) diversity (lower is better) scores. We have added a corresponding comment in the paper.
>
>
>
>
>
> * Does training SCAN with the structure provided by the logical operators lead to improved performance?
>
> We find that the logical operators do improve the diversity of samples since the training of the logical operators relies on the visual grounding that is exactly the same as SCAN uses. For example, we can recover the diversity of SCAN_R samples by training its recombination operators with a forward KL. We have added a note about this to the paper.
>
>
>
>
> * There seems to be a mistake in figure 5 unless I interpreted it incorrectly. The right side doesn't match the left side. During the middle stage of training object hues vary on the left, but floor color becomes less specific on the right. Shouldn't object color become less specific?
>
> Thank you for pointing it out. We have fixed it.
>
>
> Happy holidays!

---

> > ### Comment · AnonReviewer2 · 2018-01-01
> > **Good response**
> >
> > Thanks for the response! It nicely addressed my concerns, so I increased my rating.

---

### Decision · Program_Chairs · 2018-01-29
**ICLR 2018 Conference Acceptance Decision**

**Decision:**

Accept (Poster)

**Comment:**

This paper initially received borderline reviews. The main concern raised by all reviewers was a limited experimental evaluation (synthetic only). In rebuttal, the authors provided new results on the CelebA dataset, which turned the first reviewer positive. The AC agrees there is merit to this approach, and generally appreciates the idea of compositional concept learning.